



# A framework to evaluate and elucidate the driving mechanisms of coastal sea surface pCO$_2$ seasonality using an ocean general circulation model (MOM6-COBALT)

**Alizée Roobaert**[1], Laure Resplandy[2], Goulven G. Laruelle[1], Enhui Liao[2] and Pierre Regnier[1]

[1]Department of Geosciences, Environment & Society-BGEOSYS, Université Libre de Bruxelles, Brussels, CP160/02, Belgium
[2]Department of Geosciences, Princeton University, Princeton, NJ, USA

*Correspondence to*: Alizée Roobaert (Alizee.Roobaert@ulb.be)

## Abstract

The temporal variability of the sea surface partial pressure of CO$_2$ (pCO$_2$) and the underlying processes driving this variability are poorly understood in the coastal ocean. In this study, we tailor an existing method that quantifies the effects of thermal

changes, biological activity, ocean circulation and fresh water fluxes to examine seasonal pCO$_2$ changes in highly-variable coastal environments. We first use the Modular Ocean Model version 6 (MOM6) and biogeochemical module Carbon Ocean Biogeochemistry And Lower Trophics version 2 (COBALTv2) at a half degree resolution to simulate the coastal CO$_2$ dynamics and evaluate it against pCO$_2$ from the Surface Ocean CO$_2$ Atlas database (SOCAT) and from the continuous coastal pCO$_2$ product generated from SOCAT by a two-step neuronal network interpolation method (coastal-SOM-FFN, Laruelle et al.,

2017). The MOM6-COBALT model not only reproduces the observed spatio-temporal variability in pCO$_2$ but also in sea surface temperature, salinity, nutrients, in most coastal environments except in a few specific regions such as marginal seas. Based on this evaluation, we identify coastal regions of 'high' and 'medium' model skill where the drivers of coastal pCO$_2$ seasonal changes can be examined with reasonable confidence. Second, we apply our decomposition method in three contrasted coastal regions: an Eastern (East coast of the U.S) and a Western (the Californian Current) boundary current and a

polar coastal region (the Norwegian Basin). Results show that differences in pCO$_2$ seasonality in the three regions are controlled by the balance between ocean circulation, biological and thermal changes. Circulation controls the pCO$_2$ seasonality in the Californian Current, biological activity controls pCO$_2$ in the Norwegian Basin, while the interplay between biology, thermal and circulation changes is key in the East coast of the U.S. The refined approach presented here allows the attribution of pCO$_2$ changes with small residual biases in the coastal ocean, allowing future work on the mechanisms controlling coastal

air-sea CO$_2$ exchanges and how they are likely to be affected by future changes in sea surface temperature, hydrodynamics and biological dynamics.



## 1 Introduction

The ocean plays an important role in offsetting human-induced carbon dioxide ($CO_2$) emissions associated with cement production and fossil fuel combustion (Friedlingstein et al., 2019). Globally, the ocean is a net sink that absorbs roughly one

quarter of the anthropogenic $CO_2$ emitted into the atmosphere (-2.5 $\pm$ 0.6 Petagram of carbon per year (Pg C yr$^{-1}$) for the 2009-2018 decade, Friedlingstein et al., 2019). The spatio-temporal variability of this oceanic $CO_2$ uptake is relatively well constrained in the open ocean thanks to several method including sea surface $CO_2$ data-derived interpolations (e.g., Landschützer et al., 2014; Rödenbeck et al., 2014, 2015; Takahashi et al., 2002), models and atmospheric inversions (e.g., Gruber et al., 2009, 2019; Keeling and Manning, 2014; Manning and Keeling, 2006), but it is less constrained and understood

in the coastal ocean. Nonetheless, in recent decades, significant progress have been made with regard to the quantification and analysis of the spatial distribution of the coastal air-sea $CO_2$ exchange (FCO$_2$) globally and regionally (e.g., Borges et al., 2005; Cai, 2011; Chen et al., 2013; Laruelle et al., 2010, 2014, Roobaert et al., 2019). The FCO$_2$ seasonal cycle was also recently analyzed in coastal regions worldwide by Roobaert et al. (2019). This study identified that at the annual timescale, the global coastal ocean acts as an atmospheric $CO_2$ sink (-0.2 $\pm$ 0.02 Pg C yr$^{-1}$) with a more intense $CO_2$ uptake occurring in summer

because of the disproportionate influence of high latitude coastal seas in the Northern Hemisphere. A more in-depth analysis also revealed that the majority of the coastal seasonal FCO$_2$ variations stems from the air-sea gradient in partial pressure of $CO_2$ (pCO$_2$), although changes in wind speed and sea-ice cover can be significant regionally.

Several processes influence the seasonal variations of surface ocean pCO$_2$ and thus, the seasonality in FCO$_2$. These processes

include changes in sea surface temperature (SST) tied to air-sea heat fluxes and ocean circulation, changes in sea surface salinity (SSS) associated with evaporation, fresh water fluxes (from land, ice-melt, precipitation and evaporation) and ocean circulation, as well as variations in sea surface alkalinity (ALK) and dissolved inorganic carbon (DIC) tied to biological activity, fresh water fluxes and ocean circulation (Sarmiento and Gruber, 2006). In the open ocean, the respective influence of these processes on the pCO$_2$ variability has been interpreted using changes in SST, SSS, ALK and DIC observed in-situ (e.g.,

Landschützer et al., 2018; Takahashi et al., 1993) or based on global/regional ocean biogeochemical models relying on a mechanistic, quantitative description of the physical, chemical and biological processes controlling the ocean carbon cycle (e.g., Doney et al., 2009). These investigations reveal that changes in SST (i.e. the thermal effect) is the main driver of the seasonal pCO$_2$ in tropical oceanic regions, while non-thermal components (change associated with DIC, ALK and SSS) dominate at mid- and high-latitude (poleward of 40° N and 40° S, e.g., Landschützer et al., 2018; Takahashi et al., 2002).


In the coastal ocean, the processes controlling the pCO$_2$ seasonal dynamics was mostly investigated regionally (e.g., Arruda et al., 2015; Frankignoulle & Borges, 2001; Laruelle et al., 2014; Nakaoka et al., 2006; Shadwick et al., 2010, 2011; Signorini et al., 2013; Turi et al., 2014; Yasunaka et al., 2016) and only a few observation-based studies attempted to analyze the coastal pCO$_2$ seasonal variability into processes at the global scale (Cao et al., 2020; Chen and Hu, 2019; Laruelle et al., 2017).



Regional studies using either observations or model results have covered, e.g., the shelves of the entire Atlantic basin (Laruelle et al., 2014), the West (California Current, Turi et al., 2014) and East (e.g., Shadwick et al., 2010, 2011; Signorini et al., 2013) coasts of the United States, as well as the South and Southeast Brazilian shelves, Uruguayan and Patagonia shelves and shelves along the southwestern Atlantic ocean (Arruda et al., 2015). In the California Current, the strong upwelling of carbon-rich waters was identified as the main control of the $pCO_2$ seasonality (Turi et al. 2014). On the Patagonia shelf, the thermal effect

and biological pumps were found to be the main drivers of the seasonal $pCO_2$ variability with only a small contribution from the ocean circulation (Arruda et al., 2015), while along the East coast of the U.S, seasonal thermal changes play the major role (Shadwick et al., 2010, 2011; Laruelle et al., 2015; Signorini et al., 2013). These studies are, however, confined to specific regions and a global picture of the mechanisms driving the coastal $pCO_2$ dynamics is still missing. In addition, the attribution analysis into specific physical and biological processes is incomplete. Indeed, the attribution relies on a linear decomposition

linking variations in sea surface ocean $pCO_2$ to seasonal changes in DIC, ALK, SST and SSS (e.g., Signorini et al., 2013, Doney et al., 2009; Lovenduski et al., 2007; Takahashi et al., 1993; Turi et al., 2014), or on a series of sequential simulations isolating biological and physical terms therefore ignoring how covariations between the different terms dampen or reinforce each other (e.g., Arruda et al., 2015; Turi et al., 2014).

In this study, we develop a new framework to elucidate the seasonal $pCO_2$ dynamics of the global coastal ocean. This framework relies on the global Modular Ocean Model version 6 (MOM6, Adcroft et al., 2019) from the NOAA Geophysical Fluid Dynamics Laboratory coupled to the biogeochemical module Carbon Ocean Biogeochemistry And Lower Trophics version 2 (COBALTv2, Stock et al., 2014, 2020). MOM6-COBALT model outputs provide the relevant variables and processes that are required to perform an explicit decomposition of the inorganic carbon dynamics (Liao et al., 2020) in the

entire coastal domain. These outputs are then analyzed using a novel approach to attribute seasonal variations in surface ocean $pCO_2$ to changes in biological activity, ocean circulation, SST, air-sea $CO_2$ fluxes and fresh water fluxes (Liao et al., 2020), and which is here enhanced for the coastal ocean. The decomposition method constitutes a significant improvement upon previous studies. First, it accounts for co-variations in biological and physical processes and how their evolution jointly modulates the $pCO_2$ signal. Second, it improves on the traditional linear approaches developed for the open ocean (Sarmiento

and Gruber, 2006; Takahashi et al., 1993) and used since then (e.g. Lovenduski et al., 2007), because, as shown later in this study, the linear decomposition introducing significant biases in coastal waters due to the larger range in DIC, ALK, pH and salinity values encountered in the variable coastal environment (Egleston et al., 2010).

In light of these knowledge gaps, the objective of this paper are twofold:

-   First, we evaluate the performance of the MOM6-COBALT model in its ability to reproduce the observed spatio-temporal fields of SSS, SST, sea surface nutrients and $pCO_2$ in the global coastal domain. In particular, we identify the coastal regions where the model best reproduces the observed ocean $pCO_2$ variability and can thus be considered most suitable for a detailed analysis of the drivers of the $pCO_2$ seasonal changes.





- Second, to illustrate the capabilities of our upgraded decomposition framework, we examine the drivers of the pCO$_2$
seasonality in three contrasted coastal regions: The East coast of the U.S, the West coast of North America and the
     Norwegian Basin.

## 2 Methodology

### 2.1 Ocean biogeochemical model description

In this study, we used the ocean model MOM6 and the Sea Ice Simulator version 2 (fourth generation of ocean-ice models
called OM4) detailed in Adcroft et al. (2019). The version of OM4 adopted here is OM4p5 which has a nominal horizontal
resolution of 0.5° (i.e. with a finer latitudinal resolution of 0.26° in the tropical region). On the vertical, it includes 75 hybrid
coordinates with a z* coordinate near the surface (geopotential coordinate allowing free surface undulations) and a modified
potential density coordinate below. The vertical spacing increases from 2 m in the upper 20 m (i.e first 10 layers) to larger
isopycnal layers below. Layers in z* broadly deepens towards high latitudes (see Adcroft et al., 2019 for details on the grid).
This ocean-ice model is coupled to the biogeochemical module COBALT version 2 (COBALTv2), which includes 33 state
variables to resolve global-scale cycles of carbon, nitrogen, phosphate, silicate, iron, calcium carbonate, oxygen and lithogenic
materials (Stock et al., 2020). Details about the planktonic food web dynamics in COBALT, and global assessments of large-
scale carbon fluxes through the food web such as net primary production can be found in Stock et al. (2014, 2020). The ocean
model is forced by the 55-km horizontal resolution Japanese atmospheric reanalysis (JRA55-do) version 1.3 at a 3-hour
frequency between 1959 and 2018 (Tsujino et al., 2018), and the atmospheric pCO$_2$ data from the Earth System Research
Laboratory (Joos and Spahni, 2008). SST, SSS, sea surface nutrients (nitrate, phosphate, silicate) and oxygen were initialized
from the World Ocean Atlas version 2013 (Garcia et al., 2013a, 2013b; Locarnini et al., 2013; Zweng et al., 2013). Initial DIC
and ALK conditions are taken from GLODAPv2 (Olsen et al., 2016). The initial DIC is corrected for the accumulation of
anthropogenic carbon to match the level expected in the first year of simulation (1959) using the data-based estimate of ocean
anthropogenic carbon content of Khatiwala et al. (2013). At the end of a 81-year spin-up repeating year 1959, the model has
reached a near-equilibrium between atmospheric pCO$_2$ and surface ocean pCO$_2$, with a drift in global air-sea CO$_2$ flux < 0.004
Pg C yr$^{-1}$ over the last 10 years of spin-up. Further details on the configuration, spin-up and simulation can be found in Liao et
al. (2020).

### 2.2 Observational products and model evaluation

We first evaluate the ability of MOM6-COBALT to reproduce the observed spatial distribution of environmental variables in
the coastal domain, namely the SST, SSS and sea surface nutrients (nitrate, phosphate and silicate). The observational SST
and SSS fields are from the daily NOAA OI SST V2 (Reynolds et al., 2007) and the daily Hadley center EN4 SSS (Good et
al., 2013), respectively. The observed nutrient fields in the sea surface are extracted from the World Ocean Atlas version 2018
(Garcia et al., 2019). We also compare the simulated coastal pCO$_2$ directly to "raw", un-interpolated observations extracted





from the Surface Ocean $CO_2$ Atlas database (SOCAT), using monthly observations from SOCAT version 6 gridded at the spatial resolution of 0.25 degree (SOCATv6, Bakker et al., 2016). For the evaluation period used in this study (1998 - 2015), this database contains 9.8 million $pCO_2$ observations within the coastal domain. All data from SOCATv6 are converted from fugacity of $CO_2$ in water to $pCO_2$ using the formulation of Takahashi et al. (2012). We finally compare the $pCO_2$ simulated by the MOM6-COBALT model to the 0.25° continuous monthly $pCO_2$ fields generated from the SOCAT observations by the

two-step neuronal network (SOM-FFN) in coastal regions (Laruelle et al., 2017). The SOM-FFN data product of Laruelle et al. (2017) is thus not "raw" and implies a significant amount of statistical modelling. It is also derived from an earlier version of SOCAT (SOCATv4, Laruelle et al., 2017) than the "raw" one. In what follows, the $pCO_2$ products generated by the model, the statistical interpolation of observations, and the un-interpolated observations will be referred to as MOM6-COBALT, coastal-SOM-FFN and Socatv6, respectively. All observational and simulated fields are converted from their original spatio-

temporal resolution to monthly 0.25° gridded climatologies for the 1998 - 2015 period to match the one used by the coastal-SOM-FFN. Cells that are covered by more than 95 % of sea-ice are removed from the comparison since we assume no transfer of our master variable ($pCO_2$) through sea ice. In our analysis, we apply the broad definition of the coastal zone by Laruelle et al. (2017), using a global mask that excludes estuaries and inland water bodies while its outer limit is set 300 km away from the shoreline. This definition leads to a total surface area of 77 million km² which is split into 45 coastal regions using the

MARgins and CATchment Segmentation (MARCATS, Laruelle et al., 2013). These 45 regions are grouped into 7 broad classes with similar hydrological and climatic settings (Liu et al., 2010): (1) Eastern and (2) Western Boundary Currents (EBC and WBC respectively), (3) tropical margins, (4) subpolar and (5) polar margins, (6) marginal seas and (7) Indian margins.

The model evaluation of all gridded environmental variables including $pCO_2$ is performed for the annual mean and the seasonal

cycle both globally and within each of the 45 MARCATS regions. For the seasonal analysis, for each variable, a climatological monthly anomaly is calculated as the difference between the variable x for a given month and its climatological annual mean. The evaluation of the seasonal amplitude is then performed using the bias between observed and simulated root mean square (RMS) of their monthly anomalies. A positive bias represents a larger simulated seasonal amplitude than derived from the observations. The temporal shift between observed and simulated seasonal cycles is also assessed from the Pearson correlation

coefficient (no units) of the regression between monthly times series simulated by MOM6-COBALT and those extracted from the observations. These comparisons not only serve to assess the overall model's performance in reproducing observations but also help identifying potential discrepancies between observed and simulated environmental fields (e.g., SST, SSS) that are used by the two-step neuronal network coastal-SOM-FFN to generate the continuous $pCO_2$ climatology.

Finally, from this global and regional spatio-temporal evaluation, we label the model skill ('high', 'medium' and 'low') for each MARCATS and identify regions for which our results are the most robust for further in-depth analysis of the processes driving the coastal $pCO_2$ dynamics. The model skill labelling is based on 3 criteria: First, we assess whether the simulated annual mean $pCO_2$ is within 20 µatm of the one extracted from the coastal-SOM-FFN. This threshold of 20 µatm roughly





corresponds to the globally averaged $pCO_2$ gradient between the atmosphere and the coastal sea surface (Laruelle et al., 2018).

The second and third criteria evaluate the magnitude and phasing of the simulated $pCO_2$ seasonal cycle against the coastal-SOM-FFN, using an absolute bias in the seasonal magnitude < 20 µatm and a Pearson coefficient > 0.5 as threshold. Model skill is considered 'high' when the 3 criteria are fulfilled, 'medium' when criteria 2 and 3 are satisfied and 'low' when only one or zero criteria is met on the seasonality.

### 2.3 Processes controlling seasonal $pCO_2$ variability: a method tailored for coastal regions

$pCO_2$ in surface sea water can be computed from DIC and ALK following Eq. (1) (Sarmiento and Gruber, 2006; Wolf-Gladrow et al., 2007):

$$pCO_2 = \frac{K'_2}{K'_0 K'_1} \cdot \frac{(2DIC - ALK)^2}{ALK - DIC} \qquad (1)$$

where $K'_0$ is the aqueous-phase solubility constant of $CO_2$ in water and $K'_1$ and $K'_2$ represent the apparent equilibrium dissociation constants of the carbonate system. Several physical and biogeochemical processes can thus affect $pCO_2$ via changes in DIC, ALK and/or via the $\frac{K'_2}{K'_0 K'_1}$ term which depends on SST and SSS. To quantify the processes controlling the $pCO_2$ variability at the seasonal timescale of interest to this study, we adopt the method of Liao et al. (2020). The method starts from the traditional approach that links variations in sea surface ocean $pCO_2$ to changes in DIC, ALK, SST and SSS using the

following linear decomposition (Doney et al., 2009; Lovenduski et al., 2007; Takahashi et al., 1993; Turi et al., 2014):

$$\Delta pCO_2 \approx \frac{\partial pCO_2}{\partial DIC} \Delta DIC + \frac{\partial pCO_2}{\partial ALK} \Delta ALK + \frac{\partial pCO_2}{\partial SST} \Delta SST + \frac{\partial pCO_2}{\partial SSS} \Delta SSS \qquad (2)$$

Where the "$\Delta x$" terms represent the seasonal anomaly of x (i.e. the departure from the annual mean) and $\frac{\partial pCO_2}{\partial DIC}$, $\frac{\partial pCO_2}{\partial ALK}$, $\frac{\partial pCO_2}{\partial SST}$

and $\frac{\partial pCO_2}{\partial SSS}$ are coefficients that describe the sensitivity of $pCO_2$ to changes in DIC, ALK, SST and SSS. The coefficients for DIC, SST and SSS are always positive as $pCO_2$ increases with increases in DIC, SST or SSS, while the coefficient for ALK is always negative as $pCO_2$ systematically decreases with increasing ALK. These coefficients are generally estimated using the approach of Sarmiento and Gruber (2006) (see Eq. S1-S4 in Appendix), which has been widely used in the open ocean (Liao et al., 2020; Sarmiento and Gruber, 2006; Takahashi et al., 1993). In this study, we refine the estimation of the coefficients so

they can be used for the wide range of DIC/ALK ratios that can be encountered in the coastal waters. This includes conditions when the DIC/ALK ratio is close to 1, such as in regions with significant freshwater discharge like those found near estuarine mouths or on polar shelves subject to sea-ice melting, when pH is around 7.5 (Egleston et al., 2010). In these case, the traditional approximation method breaks down (see Eq. (S1-S2) and Figure S1 in the Appendix). To circumvent this important limitation, we computed the coefficients of the $pCO_2$ dependency using a regression approach based on the CO2SYS program



(Lewis and Wallace, 1998). At each point in space, $pCO_2$ was computed using the 1998 - 2015 average of DIC, ALK, SSS and

SST with CO2SYS (method 14 in CO2SYS Matlab program, Millero, 2010). The $\frac{\partial pCO_2}{\partial DIC}$ coefficient was then computed as the

slope of the linear regression between $pCO_2$ and DIC obtained by allowing DIC to vary around the local mean DIC value while

keeping other tracers (ALK, SST, SSS) constant. The DIC range used to compute the slope was set to the $\pm 2$ standard deviation

of the 1998-2015 monthly values at that location with an upper bound at $\pm 60$ µmol kg$^{-1}$ (see Appendix for further details).

The same approach was repeated to compute the coefficients for the $pCO_2$ dependence on ALK, SST and SSS, respectively.

Our methodology leads to coefficients that are constant in time but space dependent. In Fig. S1, we compare the coastal $pCO_2$

reconstructed from the traditional decomposition (using the empirical coefficients reported by Sarmiento and Gruber, 2006)

with those computed here. For the global coastal ocean, we find a large bias (global mean rmse of fitting $pCO_2$ anomaly in Eq.

(2) = 14.6 µatm), which is especially pronounced at high latitudes. In contrast, the decomposition method based on our

methodology reduce drastically the biases (global mean rmse = 2.8 µatm) in coastal regions and allows a more robust

reconstruction of the $pCO_2$ variability.

Here we assume that the coefficients are constant in time, and the temporal change in $pCO_2$ ($\partial_t pCO_2$ in µatm month$^{-1}$) can

therefore be expressed as a simple function of the temporal changes in DIC ($\partial_t DIC$), ALK ($\partial_t ALK$), SST ($\partial_t SST$) and SSS

($\partial_t SSS$):

$$\partial_t pCO_2 \approx \frac{\partial pCO_2}{\partial DIC}\partial_t DIC + \frac{\partial pCO_2}{\partial ALK}\partial_t ALK + \frac{\partial pCO_2}{\partial SST}\partial_t SST + \frac{\partial pCO_2}{\partial SSS}\partial_t SSS \tag{3}$$

Temporal changes in DIC, ALK, SST, and SSS ($\partial_t DIC$, $\partial_t ALK$, $\partial_t SST$ and $\partial_t SSS$) are controlled by surface heat flux, ocean

transport, freshwater fluxes, biological processes, and the air-sea $CO_2$ flux. Using the model results, we further expand the

decomposition to quantify the contribution of these physical and biological processes (see details of derivation in Liao et al,

2020) :

$$\underbrace{\partial_t pCO_2}_{pCO_2\ change} \approx$$

$$\underbrace{\left(\frac{\partial pCO_2}{\partial DIC}\partial_t DIC_h + \frac{\partial pCO_2}{\partial ALK}\partial_t ALK_h + \frac{\partial pCO_2}{\partial SSS}\partial_t SSS_h + \frac{\partial pCO_2}{\partial DIC}\partial_t DIC_v + \frac{\partial pCO_2}{\partial ALK}\partial_t ALK_v + \frac{\partial pCO_2}{\partial SSS}\partial_t SSS_v\right)}_{circ}$$

$$+ \underbrace{\left(\frac{\partial pCO_2}{\partial DIC}\partial_t DIC_{fw} + \frac{\partial pCO_2}{\partial ALK}\partial_t ALK_{fw} + \frac{\partial pCO_2}{\partial SSS}\partial_t SSS_{fw}\right.}_{fw}$$

$$+ \underbrace{\left(\frac{\partial pCO_2}{\partial DIC}\partial_t DIC_{bio} + \frac{\partial pCO_2}{\partial ALK}\partial_t ALK_{bio}\right)}_{bio}$$





$$+ (\underbrace{\frac{\partial pCO_2}{\partial T}(\partial_t SST_h + \partial_t SST_v + \partial_t SST_q)}_{thermal}$$

$$+ (\underbrace{\frac{\partial pCO_2}{\partial DIC}\partial_t DIC_{CO_2\ flux})}_{CO_2\ flux} \tag{4}$$


where the temporal changes in pCO₂ (time tendency called *pCO₂ change*) is on the left-hand side (LHS), and the five terms that control this change in pCO₂ are on the right-hand side (RHS) of the equation. Subscripts *h* and *v* denote the contribution from horizontal (advection and diffusivity in the meridional and zonal directions) and vertical (vertical advection and diffusivity) transports on SST, SSS, DIC and ALK, *bio* denotes the DIC and ALK changes induced by biological processes
(photosynthesis, respiration, and calcium carbonate dissolution/precipitation, denitrification and nitrification), *q* denotes the effect of surface heat flux on SST, *fw* denotes the effect of fresh water fluxes (i.e., precipitation, evaporation, river runoff and sea-ice formation and melting) on SSS, DIC and ALK, and the term *CO₂ flux* denotes the DIC change induced by air-sea CO₂ exchange.

Here we examine changes in pCO₂ attributed to three oceanic processes that modify the concentration in dissolved species (i.e. DIC, ALK and SSS), namely their transport by oceanic circulation (*circ*, which include horizontal and vertical transport), the effect of dilution/concentration due to freshwater fluxes (*fw*) and the effect of biological activity (*bio*), and isolate the *thermal* influence tied to SST changes induced by both oceanic transport and air-sea exchange of heat. Finally, the air-sea CO₂ exchange (*CO₂ flux*) pushes the surface pCO₂ concertation towards its equilibrium with the atmosphere and systematically acts to offset
the pCO₂ changes associated with the sum of the internal oceanic processes (*circ*, *bio*, *fw* and *thermal*). In this study, we apply Eq. (4) using averages between the sea surface and the mixed layer depth (MLD), defined here as the depth where the water density is 0.01 kg m⁻³ denser than the water at the surface (minimum MLD is 5 meters). Positive contributions on the RHS would yield an increase in pCO₂ (positive pCO₂ response on the LHS). Positive values of the *CO₂ flux* correspond to an ocean CO₂ uptake. This method to decompose the pCO₂ seasonality into controlling processes in the coastal domain is illustrated in
three coastal regions: The East and West coast of North America and in the Norwegian Basin.

## 3 Results and discussion

### 3.1 Annual mean state and seasonal cycle model evaluation and identification of coastal regions

Figure 1a identifies the coastal regions where the performance of MOM6-COBALT is satisfactory for both the annual mean and the seasonal cycle of pCO₂. The analysis, performed at the MARCATS scale (see Fig. 1b for nomenclature), distinguishes
regions of low, medium and high model skill, the latter being areas for which our confidence in the identification of the dominant biophysical drivers of the coastal pCO₂ dynamics is highest. This figure will be analyzed in detail in Section 3.1.3, but before we do so we first perform a data-model evaluation according to the following: We first evaluate the model by





comparing simulated fields of SSS, SST, sea surface nutrients to observations globally and regionally (Sect. 3.1.1, Figs. 2 and 3). Second, the ability of the model to capture the coastal $pCO_2$ annual mean and seasonality is assessed against the "raw"

Socatv6 data and the continuous monthly observational-based $pCO_2$ product (coastal-SOM-FFN, Laruelle et al., 2017), respectively (Sect. 3.1.2, Figs. 3-6).

### 3.1.1 Model evaluation for coastal waters environmental variables

MOM6-COBALT captures fairly well the main spatial patterns of key environmental parameters (SST, SSS and sea surface nutrients) in the coastal domain (Fig. 2). The global SST field simulated by the model reproduces the strong large-scale tropical

to polar SST gradients, with a global median bias of -0.2 °C (Fig. 2a-c), and biases at the scale of MARCATS regions ranging from 0 °C in the North East Atlantic (M17) to 1.3 °C in the East coast of the U.S (M10, Fig. 3a and Table S1). With a global median bias value of 0.2, the model also correctly reproduces the observed SSS patterns which are mainly regulated by evaporation and freshwater inputs from precipitation, riverine runoff and ice melt, with lower SSS values in polar regions and along the coasts in Southeast Asia and higher SSS values along the coasts of evaporation basins such as in the Arabian or the

Mediterranean Sea (Fig. 2d-f). The SSS analysis at the MARCATS scale reveals absolute SSS biases generally less than or close to 1 except for five MARCATS where absolute biases exceed 2. These MARCATS are mainly located in marginal seas (the Baltic Sea, M18, the Black Sea, M21 and the Persian Gulf, M29), but also include one polar region (the Canadian Archipelago, M13) and one tropical region (Tropical West Atlantic, M7, Fig. 3b and Table S1). Similar to SSS, largest model-data discrepancies for nutrients are mostly found in marginal seas (Fig. 3c-e and Table S1). For instance, the largest $PO_4$ and

$SiO_4$ biases are encountered in the Black Sea (M21, absolute biases of 3 and 75 µmol kg$^{-1}$, respectively). The Peruvian upwelling (M4), the Bay of Bengal (M31) and the N-E Pacific (M1) also present large biases in $NO_3$ and $PO_4$, respectively (e.g., $NO_3$ bias of 8 µmol kg$^{-1}$ for M4). The global median nutrients biases are however much smaller, reaching 0.3, -0.2 and -0.4 µmol kg$^{-1}$ for nitrate ($NO_3$, Fig. 2i), phosphate ($PO_4$, Fig. 2l) and silicate ($SiO_4$, Fig. 2o), respectively,

The model-data seasonal evaluation reveals that MOM6-COBALT reproduces the global SST and SSS amplitudes remarkably

well (median absolute bias of 0.1 °C and 0.0, respectively, Table S2). Some exceptions can nevertheless be diagnosed such as in the marginal Black Sea (M21) where the bias in SST seasonal amplitude reaches -1.3 °C, and in three MARCATS (The Bay of Bengal (M31), the Tropical West Atlantic (M7), and the Siberian Shelves (M43)) where the SSS seasonal biases are larger than 0.4. The model-data comparison also reveals that the phasing of the SST and SSS seasonal cycles are in very good agreement (Pearson correlation close to 1) for all 45 MARCATS but four, for which significant deviations in SSS are found:

two marginal seas (Hudson Bay, M12 and the Red Sea, M28) and along the Californian (M2) and Brazilian Currents (M6). The nutrients analysis shows absolute global median biases in seasonal amplitude of 0.1, 0.0 and 0.7 µmol kg$^{-1}$ for $NO_3$, $PO_4$ and $SiO_4$, respectively. Seven MARCATS present absolute biases larger than 1.5 µmol kg$^{-1}$ mainly located in marginal seas (Baltic Sea, M18 and the seas of Japan (M40) and Okhotsk (M41)), but also in polar (Siberian (M43) and Antarctic (M45) shelves) and subpolar (N-E Pacific, M1) regions and in the Bay of Bengal (M31). The model-data comparison sometimes





shows significant phases shift in their seasonal signal (Pearson coefficient < 0.5), such as for MARCATS located in Indian and Tropical margins, marginal seas and EBCs.

### 3.1.2 Model evaluation for coastal pCO₂

The spatial distribution of the annual mean $pCO_2$ simulated by MOM6-COBALT is in good agreement with the observational $pCO_2$ values extracted from the "raw" Socatv6 database with generally low $pCO_2$ values (blue colors) in temperate and high

latitudes and high $pCO_2$ values (yellow and red colors) in tropical and sub-tropical regions (contrast Fig. 4a with Fig. 4b). The model-data $pCO_2$ evaluation at the regional scale shows that 33 of the 45 MARCATS present absolute biases lower than 20 µatm (Table S1). The regions where the bias exceeds this threshold include two EBC's (M2 and M4), two marginal seas (M40 and M41), and one Polar (M45), subpolar (M42) and Tropical East Atlantic (M23) shelf. Note that in some MARCATS regions, in particular in marginal seas and Indian seas, there are no Socatv6 observations to perform the comparison (e.g. the Bay of

Bengal, M31, see Fig. 4b and Table S1). Hence, we also evaluate the performance of MOM6-COBALT against the continuous coastal-SOM-FFN $pCO_2$ product which uses a neural network interpolation method to fill data gaps and resolve the spatio-temporal coastal $pCO_2$ variability globally.

Our results show that MOM6-COBALT reproduces the main spatial features of the annual mean $pCO_2$ field captured by the coastal-SOM-FFN product, as revealed by the relatively low globally averaged bias of 2.5 µatm (Figs. 4a and 4c). In both the

model and the SOM-FFN product, low coastal $pCO_2$ values are consistently found in temperate and high latitude regions in both hemispheres, while high $pCO_2$ values are largely limited to (sub)tropical regions. Largest discrepancies (Fig. 4d) are found at high latitudes (poleward of 60° N and 60° S, negative bias), along the Eastern Boundary Peruvian and Namibian upwelling systems (high positive bias) and more locally close to the mouth of some large rivers (e.g., the plume of the Amazon or the Rio de la Plata, high negative bias). We note however that these regions are poorly sampled in the Socatv6 dataset (Fig.

4b) and are thus likely weakly constrained in the coastal-SOM-FFN product (Fig. 4c).

At the regional scale, differences in annual mean $pCO_2$ between MOM6-COBALT and coastal-SOM-FFN are lower than 20 µatm in 35 MARCATS (Table S1, Fig. 3f), which partly is a reflection of the low annual mean biases observed in the environmental driver variables in these regions (see Sect. 3.1.1). In EBC, WBC, and subpolar coastal regions, the model tends to overestimate the regional mean $pCO_2$ compared to coastal-SOM-FFN (positive bias), except along the East coast of U.S

(M10), in the China and Kuroshio seas (M39) and in the North East Atlantic (M17, Table S1). In polar regions, the model generally underestimates the mean $pCO_2$ compared to coastal-SOM-FFN, except around the South of Greenland (M15). In Indian, marginal, and tropical coastal regions, no general trend can be identified regarding the sign of the bias, which can be positive or negative.



Quantitatively, the 10 MARCATS with absolute biases > 20 µatm are mainly located in regions for which very limited or no
observational data have been compiled in the Socatv6 database (Table S1) and/or for which large discrepancies can already be
identified at the level of the master environmental variables (Sect. 3.1.1). These regions belong mainly to EBCs (3 out of the
6 EBC MARCATS), marginal seas (3 out of the 9 marginal seas MARCATS), the remaining four being either polar (M13 and
M14), subpolar (M42) or Indian margins (M31). The largest biases are found in the Peruvian upwelling Current (M4), the
South West of Africa (M24), the Californian upwelling Current (M2) and the Canadian Archipelago (M13) with biases of 106
µatm, 79 µatm, 35 µatm and -53 µatm, respectively.

Our analysis reveals that the seasonal amplitudes simulated by MOM6-COBALT are systematically larger than the ones
estimated by the coastal-SOM-FFN product (Fig. 5a-b, red colors in Fig. 5c and positive biases in Table S2) for all coastal
regions belonging to EBC, WBC, Indian and tropical margins. For the majority of polar and subpolar margins and for some
marginal seas, the model simulates lower seasonal $pCO_2$ amplitudes (blue colors in Fig. 5c and negative biases in Table S2).
Note that the seasonal evaluation is only performed against the coastal-SOM-FFN product because only few 0.25° coastal cells
(approximately 45) in the Socatv6 database contain complete continuous $pCO_2$ time series. Quantitatively, absolute biases
between the modelled and coastal-SOM-FFN amplitudes do not exceed 20 µatm except for marginal seas where larger
discrepancies are calculated (6 of the 9 marginal MARCATS, Table S2). The monthly mean $pCO_2$ seasonal cycle simulated
by MOM6-COBALT is also well in phase (Pearson correlation coefficients > 0.5) with the one extracted from coastal-SOM-
FFN in 34 out of the 45 MARCATS (red colors in Fig. 5d and Table S2). The agreement is especially good in the best monitored
MARCATS regions (MARCATS where > 50 % of the area is covered by Socatv6 observations, Table S1). For instance, in
regions with good data coverage such as along the East coast of the U.S (M10, Fig. 6a), the Norwegian Basin (M16, Fig. 6b),
the Californian Current (M2, Fig. 6c), the Leeuwin Current (M33), or the Brazilian Current (M6), the Pearson correlation
coefficient is higher than 0.9 (Table S2). In contrast, the seasonal $pCO_2$ cycle simulated by MOM6-COBALT substantially
diverges from that of the coastal-SOM-FFN in four poorly monitored marginal seas (M12, M21, M28, M29) and in a few of
the EBCs, Indian margins, subpolar margins (e.g., New Zealand, Fig. 6d) and tropical margins (Pearson correlation coefficient
< 0.5, Table S2 and blue colors in Fig. 5d).

### 3.1.3 Identifying coastal regions of 'high' model skill

Overall, the $pCO_2$ spatio-temporal analysis model-data evaluation shows that out of 45 MARCATS, 29 have an absolute bias
for their annual mean < 20 µatm when MOM6-COBALT-coastal-SOM-FFN, MOM6-COBALT-Socatv6 and coastal-SOM-
FFN-Socatv6 are compared (Table S1). Together, these 29 MARCTAS represent 65 % of the global coastal ocean surface
area. For the 11 MARCATS that are best covered by observations (MARCATS where > 50 % of the surface area is covered
by Socatv6 observations, Table S1), absolute biases for the annual mean are always < 20 µatm for the three product
intercomparison, except in the Californian Current (M2), in the Baltic Sea (M18) and along the N-E Pacific (M1). The seasonal





MOM6-COBALT against coastal-SOM-FFN evaluation also reveal that 39 of the 45 MARCATS have $pCO_2$ seasonal amplitude biases < 20 µatm and 34 MARCATS have a Pearson correlation coefficient > 0.5 (Table S2).

Based on this evaluation, we attribute for each MARCATS a level of confidence on the model skill ('high', 'medium' and 'low', Table 1 and Fig. 1a). Out of the 45 MARCATS, 25 are labeled as 'high' skill, that is to say, they fulfil the following
criteria regarding the annual mean and the seasonality (Table 1 and dotted MARCATS regions in Fig. 1a): a bias < 20 µatm in the annual mean $pCO_2$ between MOM6-COBALT and coastal-SOM-FFN, a bias < 20 µatm in the magnitude of the seasonal $pCO_2$ cycle and a seasonal phase characterized by a Pearson correlation coefficient > 0.5. Note that these MARCATS but M43, M45, M1, M23 and M26 also present an annual mean $pCO_2$ bias < 20 µatm in the MOM6-COBALT-Socatv6 and coastal-SOM-FFN-Socatv6 comparisons (Table S1). In addition, 7 'high skill' MARCATS also show a data density > 50 % (13
MARCATS if we lower the data coverage to > 30 %, Fig. 1a). These 7 MARCATS are located in contrasted coastal environments, i.e. 3 EBCs (Iberian (M19) and Moroccan (M22) upwellings and the Leeuwin Current, M33), 1 WBC (East coast of the U.S, M10), 1 Polar (Norwegian Basin, M16), 1 subpolar (NE Atlantic, M17) and 1 marginal sea (Gulf of Mexico, M9). These 7 'high' skill MARCATS could also result from the very good correspondence on the data-model annual mean and seasonal patterns in environmental fields (Table S1 and Table S2 except M22, M33 and M9 for the nutrient phasing) and
are therefore excellent potential candidates for an analysis of the processes controlling the coastal $pCO_2$ dynamics. 6 additional MARCATS regions fulfil the criteria related to the seasonal $pCO_2$ evaluation while they fail to fulfil the annual mean $pCO_2$ bias threshold of 20 µatm. These 'medium' skill regions (Table 1 and dashed regions in Fig. 1a) include 2 EBCs (Californian Current, M2 and SW Africa, M24), 1 marginal sea (Sea of Okhotsk, M41), 2 polar (Canadian Archipelago M13 and N Greenland, M14) and 1 subpolar (NW Pacific, M42) shelves. The majority of marginal seas are systematically associated with
large biases whether on the $pCO_2$ or on the main environmental variables. These regions fulfill only one or no criteria regarding the $pCO_2$ seasonality, and they are hence labeled as 'low' skill (Table 1, Fig. 1a). Other 'low' skill regions include 1 EBC (Peruvian upwelling Current, M4), 1 Indian (Bay of Bengal, M31), 2 tropical (Tropical E Pacific, M3 and SE Asia, M38), 2 subpolar (Sea of Labrador, M11 and New Zealand, M36) and 1 WBC (Brazilian Current, M6) margins.

### 3.1.4 Methodological limitations

While our results show a relatively good agreement between MOM6-COBALT and coastal-SOM-FFN regarding the spatial and temporal $pCO_2$ distribution over the global coastal ocean, the comparison remains challenging for several reasons.

First, while the climatology of Laruelle et al. (2017, coastal-SOM-FFN) is currently the best available product for a model-data comparison, it has its own limitations. For instance, in some regions, particularly coastal upwellings such as the Moroccan
(M22) and Peruvian (M4) upwellings, the $pCO_2$ fields generated by the coastal-SOM-FFN do not reproduce well the high and variable $pCO_2$ values measured in-situ (see e.g., Friederich et al., 2008 and McGregor et al., 2007). Such poor performance of the coastal-SOM-FFN algorithm in these types of systems were already identified by Laruelle et al. (2017). Indeed, upwelling





regions are still relatively poorly monitored and expand partly beyond the coastal domain used by Laruelle et al. (2017), leading

to locally skewed calibration of the SOM-FFN. Deficiencies in the observation-based product can thus partly explain the large

model-data bias (106 μatm, largest of all MARCATS) calculated in the Peruvian upwelling region. Moreover, although the

Surface Ocean $CO_2$ Atlas database (SOCAT) has expanded significantly over the past few years, some regions are still poorly

monitored. In the coastal regions where no observational data exist (e.g., in the Black Sea, the Sea of Okhotsk, the Bay of

Bengal, Fig. 4b) in the SOCAT database used here (SOCATv6, Bakker et al., 2016), it is difficult to evaluate the performance

of the SOM-FFN and, thus, of an OGCM in reproducing the $pCO_2$ field. In addition, for certain regions subjected to complex

dynamic biogeochemical settings (e.g., upwelling, seasonal cover of sea-ice, influenced by rivers, marginal seas), the $pCO_2$

field reconstructed by the SOM-FFN suffers from poor performance, which can partly be explained by the lack of observational

data. This lack of observations could partly explain why MOM6-COBALT-coastal-SOM-FFN $pCO_2$ biases exceed 20 μatm

in these regions. This study highlights the regions (Fig.1a, e.g., Indian ocean margins, Peruvian upwelling, marginal seas)

where new observational data are most urgently needed to improve our understanding of the $CO_2$ exchange between coastal

regions and the atmosphere at the regional and global scales. In addition, only one global continuous $pCO_2$ climatology derived

by the SOM-FFN method currently exists for the coastal ocean. It would therefore be beneficial for the community to develop

other observation-based climatologies relying on other interpolation techniques, as currently the case for the open ocean.

Second, the model-data comparison should also be analyzed in the light of the current limitations in the model itself. OGCMs

have been designed for global ocean applications and the coarse spatial resolution of these models, on the order of 0.5° in the

present study, cannot resolve accurately mesoscale and sub-mesoscale processes as well as tidal mixing in shelf regions even

with a model configuration including parameterizations for these processes. The coastal currents are also not always well

resolved because of the coarse resolution of the shelf bathymetry. These small-scale hydrodynamic features are known to affect

the spatio-temporal variability of $pCO_2$ and the air-sea $CO_2$ exchange (Bourgeois et al., 2016; Kelley et al., 1971; Lachkar et

al., 2007; Laruelle et al., 2010). Therefore, although MOM6-COBALT runs at 0.5°, discrepancies between coastal-SOM-FFN

and MOM6-COBALT in narrow EBCs such as the Peruvian Upwelling Current (M4) and along South west Africa (M33)

could also be explained by the limited spatial resolution of the model. Moreover, OGCMs such as MOM6-COBALT have a

relatively simple representation of the biogeochemistry which does not fully captures some of the important processes of the

carbon dynamics in coastal waters such as sea-ice temporal dynamics (Adcroft et al., 2019), neritic calcification (O'Mara and

Dunne, 2019), or terrestrial and marine organic matter decomposition and burial (Lacroix et al., 2021). Moreover, the largest

biases observed in marginal seas can partly be explained by large fluvial inputs and oceanic water flows through fine scale

topography (e.g. straits) that are poorly represented in global OGCMs.

Finally, the annual mean/seasonal $pCO_2$ biases between the coastal-SOM-FFN and MOM6-COBALT can also be traced back

to divergences in the environmental fields simulated by the model compared to observations (Table S1 and Table S2). For

instance, in most marginal seas, the model poorly resolves the annual mean and seasonal cycle of SSS and nutrients compared



to the observations. These discrepancies impact the simulated $pCO_2$ via the controls of the SSS on the $CO_2$ solubility and of nutrients on the biological pump and $CO_2$ uptake. In the tropical W Atlantic (M7) which is under the influence of the Amazon River, the model simulates lower annual mean SSS (and therefore lower $pCO_2$) than the observations. In the tropical E Pacific
(M3) and in South-East Asia (M38), the poor agreement between simulated and observed seasonal $pCO_2$ cycle could be explained by significant biases in the nutrient seasonal cycles (low Pearson correlation coefficient). Interestingly however, some regions reveal significant biases in the major environmental fields but not in the $pCO_2$ (e.g., Tropical W Atlantic, M7) while in other regions, the reverse is observed (e.g., M20, M27 and M36). Also, for some regions biases in environmental fields do not affect the $pCO_2$ as expected. For instance, along the East coast of the U.S (M10), MOM6-COBALT simulates
larger SST compared to observations while the simulated $pCO_2$ is lower compared to coastal-SOM-FFN on an annual mean. This clearly shows that biases in environmental fields are not sufficient to explain fully the biases in $pCO_2$ diagnosed between MOM6-COBALT and coastal-SOM-FFN.

## 3.2 Processes governing the seasonal $pCO_2$ variability

Our second objective is to examine the drivers of the $pCO_2$ seasonality in three well sampled and contrasted coastal regions
where the model skill is satisfactory: The East coast of North America (M10), the Norwegian Basin (M16) and the Californian Current (M2). The East coast of North America is a sink of atmospheric $CO_2$ that has been extensively studied over the past decade (e.g., Fennel et al., 2019; Laruelle et al., 2015; Shadwick et al., 2010, 2011; Signorini et al., 2013). The $pCO_2$ spatio-temporal dynamics in this MARCATS is particularly well captured by MOM6-COBALT ('high' skill, Fig. 1a), despite an annual mean SST bias of 1.3 °C on the data-model comparison in this region (Table S1). Because the SST amplitude and
seasonal phasing are in agreement between the model and data (Table S2), the bias on the mean SST does not impact the seasonal $pCO_2$ cycle (Pearson correlation coefficient > 0.5 and bias < 20 µatm on the seasonal $pCO_2$ amplitude, Table 1). We also selected the Californian Current because it is a source of $CO_2$ to the atmosphere, and similarly to the East coast of the U.S, it ranks among one of the best monitored coastal regions in the world (e.g., Evans et al., 2011; Fennel et al., 2019; Hales et al., 2012; Turi et al., 2014). In this region, the model is classified as 'medium' skill (Table 1 and Fig. 1a). Indeed, the simulated
seasonal cycle of $pCO_2$ is in relatively good agreement with coastal-SOM-FFN (Figs. 5-6, and Table 1), despite biases in the annual mean $pCO_2$ compared to observations (Fig. 3f) and a phase shift in the seasonality of SSS and nutrients (Pearson correlation coefficient < 0.5). However, the Californian Current is also one of the few coastal regions where an analysis of the processes controlling the $pCO_2$ seasonality has already been performed using a regional biogeochemical model and sequential simulation removing processes one after the other (Turi et al., 2014), which can hence be compared to our analysis. Finally,
the choice of the Norwegian Basin is motivated by the good performance ('high' skill) of the model and the intense atmospheric $CO_2$ sink that occurs in this contrasted region.



### 3.2.1 Seasonality along the East coast of North America

The seasonal evolution of pCO$_2$ averaged over the East coast of the U.S (M10) is represented in Fig. 7a. Ocean pCO$_2$ is minimum in winter (February/March ~ 331 µatm), it increases through spring and peaks in summer (August, ~ 400 µatm)
before decreasing again in the fall. Figure 7b reveals the complex interplay of the four ocean internal processes (thermal, biology, ocean circulation, and fresh water flux) on the seasonal pCO$_2$ variability which can either act in synergy or oppose each other.

The thermal effect (*thermal*, red line on Fig. 7b) increases pCO$_2$ from early spring to summer by decreasing the solubility of
CO$_2$. In contrast, the solubility of CO$_2$ increases in autumn and winter, inducing a decline in pCO$_2$. The largest changes in pCO$_2$ associated with the change in SST occur during spring (29 µatm month$^{-1}$ in June) and fall (-26 µatm month$^{-1}$ in November). This thermal effect was already identified by Signorini et al. (2013) in their observational study. These authors highlighted that lowest pCO$_2$ was generally reported in winter or at the beginning of spring and highest pCO$_2$ in summer or autumn, despite significant temporal and spatial heterogeneity between the different sub-regions of the East coast of the U.S
(Scotian shelf, the Gulf of Maine, the Georges Bank/Nantucket shoals, the Middle Atlantic Bight, and the South Atlantic Bight). The effect of biology above the mixed layer depth (*bio*, green line) reduces pCO$_2$ throughout the year revealing that primary production exceeds organic matter degradation in the surface layer all year long. The largest pCO$_2$ decrease associated with biology is observed in early spring (values of -68 µatm month$^{-1}$ in April) which is well documented (e.g., Shadwick et al., 2010, 2011; Signorini et al., 2013). The transport of chemical species by ocean circulation (*circ*, blue line) increases pCO$_2$
and tends to oppose biology year-round except at the end of fall/beginning of winter. This pCO$_2$ increase induced by the circulation term is maximum in April (26 µatm month$^{-1}$). Throughout the year, the contribution of fresh water fluxes (*fw*, pink line) remains minor compared to the other terms (maximum absolute value of 9 µatm month$^{-1}$ in January). For each month/season, the air-sea CO$_2$ exchange term (*CO$_2$ flux*, black line) counteracts change in pCO$_2$ associated with ocean internal processes taking place in surface seawater (sum of *bio*, *circ*, *thermal* and *fw*). The *CO$_2$ flux* term increases pCO$_2$ at the sea-
surface (acting as an atmospheric CO$_2$ sink) throughout the year except during summer (between July and September) where it decreases sea surface pCO$_2$ and releases CO$_2$ towards the atmosphere (acting as an atmospheric CO$_2$ source). This simulated atmospheric CO$_2$ uptake all year long except for the summer season is also in agreement with previous literature (Fennel et al., 2019; Laruelle et al., 2015; Signorini et al., 2013). The study of Laruelle et al. (2015) has nevertheless shown that in spring, the southern part of the Eastern North American coast is quasi neutral and that in fall, some regions such as the Gulf of Maine
or the Georges Bank acts as a CO$_2$ source. The temporal change of pCO$_2$ (*pCO$_2$ change*, cyan line) is the result of the non-perfect balance between the internal processes and the air-sea CO$_2$ flux.

We evaluate the rate of change tied to each process during the marked peak-to-peak pCO$_2$ increase observed between winter and summer (from 331 µatm in February to 400 µatm in August, Fig. 7a). A positive rate of change (in µatm month$^{-1}$) indicates





475 that the process contributes to an increase in pCO$_2$ between winter and summer (February-August). This process-based analysis reveals that the winter-to-summer pCO$_2$ increase in the East coast of the U.S (M10) mainly results from thermal (rate of change = +5 µatm month$^{-1}$) and ocean circulation (rate of change = +4 µatm month$^{-1}$) influences combined with a large reduction of the biological CO$_2$ uptake (rate change of +7 µatm month$^{-1}$, Fig. 7b). The importance of the thermal and circulation effects as well as the presence of a strong biological drawdown are in line with results from past studies (e.g., Laruelle et al. (2015),

480 Shadwick et al. (2010, 2011) and Signorini et al. (2013)). Our results which identifies the reduction of biological carbon uptake as a key control of pCO$_2$ seasonality agree with the studies of Shadwick et al. (2010, 2011), but slightly diverge compared to those of Signorini et al. (2013) or Laruelle et al. (2015), which found that the thermal effect was the dominant driver. This difference is largely explained by the different levels of details in the decomposition method. While most model studies, including ours, use seasonal change in SST, SSS, DIC and ALK, observational approaches cannot isolate the compounding

485 changes tied to biological activity from those of ocean transport.

### 3.2.2 Seasonality in the Norwegian basin and in the Californian Current

The pCO$_2$ seasonal cycle in the Norwegian Basin (M16) and the Californian Current (M2) simulated by MOM6-COBALT are represented in Fig. 7c and Fig. 7e, respectively. The Norwegian Basin shows a near-constant pCO$_2$ value (~ 330 µatm) throughout the year except in spring when it drops by 30 µatm (minimum pCO$_2$ value of 300 µatm in June). The phasing of

490 the seasonal pCO$_2$ cycle in the Californian Current is similar to that along the East coast of U.S, with a minimum pCO$_2$ value of 366 µatm in March followed by an increase that reaches a maximum pCO$_2$ value of 433 µatm in August and then decreases again at the beginning of the fall.

The decomposition of the seasonal cycle into different processes for both the Norwegian Basin and the Californian Current

495 (Fig. 7d and Fig. 7f) reveal patterns that are qualitatively similar to those already diagnosed for the East coast of the U.S (Fig. 7b). For both shelf regions, the biological and circulation effects respectively remain negative and positive throughout the year, while the thermal effect increases pCO$_2$ in spring and summer but decreases pCO$_2$ in fall and winter. The fresh water term is also minor compared to the other terms. Quantitatively, however, the amplitude of the different terms points to different first order control in the pCO$_2$ seasonality for each region. The amplitudes are calculated here using the marked peak-to-peak

500 change in pCO$_2$ which occurs between February and June in the Norwegian basin and between March and August in the Californian Current.

In the Norwegian basin, the strong winter to summer pCO$_2$ decreases (43 µatm, Fig. 7c) is mainly associated with the large and rapid CO$_2$ uptake associated with the spring phytoplankton bloom (biological rate of change = -45 µatm month$^{-1}$ in average

505 between February and June and with a maximum pCO$_2$ uptake of -175 µatm month$^{-1}$ in June, Fig. 7d). This biological drawdown is only partly compensated by the supply of high pCO$_2$ water masses by the ocean circulation (rate of change = +24 µatm month$^{-1}$). This dynamics is consistent with the fact that the Norwegian Basin is one of the most productive region of the





world characterized by a well-documented, intense spring bloom (e.g., Findlay et al., 2008). In addition, the effect of thermal changes only plays a comparatively minor role here (rate of change = +7 µatm month$^{-1}$).


In contrast to the East coast of the U.S and the Norwegian Basin, the analysis performed in the Californian Current reveals that circulation is the main driver of the winter-to-summer $pCO_2$ increases (68 µatm, Fig 7e). The upwelling of high-$pCO_2$ waters increases surface $pCO_2$ year-round. Its influence is however weaker in winter than in summer, thereby explaining the $pCO_2$ increase observed between February and August (rate of change = +12 µatm month$^{-1}$, Fig. 7f). This large contribution

from circulation is consistent with the simulations of Turi et al. (2014), which identified the ocean transport associated with upwelling in the Californian Current as the dominant process, and the higher intensity of the summer upwelling and its impact on $pCO_2$ was also reported in prior work (e.g., Evans et al., 2015; Fiechter et al., 2014; Turi et al., 2014). In this region, biology also opposes the effect of ocean circulation, with upwelled deep water bringing nutrients to the surface and stimulating phytoplankton productivity (e.g., Evans et al., 2015; Fiechter et al., 2014; Turi et al., 2014). However, it plays a minor role in

the $pCO_2$ increase (rate of change ~ 0 µatm month$^{-1}$) as well as for the thermal effect (rate of change = +4 µatm month$^{-1}$).

**4 Conclusions**

In this study, an OGCM (MOM6-COBALT) which is primarily designed for the open ocean was used to examined sea surface $pCO_2$ seasonality in the coastal domain. We first evaluated the ability of the model to reproduce the spatial and temporal dynamics of key environmental variables, such as SST, SSS and sea surface nutrients against in-situ observations. The spatio-

temporal variability of coastal $pCO_2$ was also evaluated using direct coastal $pCO_2$ observations from the SOCAT database (Socatv6, Bakker et al., 2016), and a global observational continuous monthly $pCO_2$ climatology available at high spatial resolution (coastal-SOM-FFN, Laruelle et al., 2017).

Our model-data comparison showed a relatively good agreement on the environmental variables spatio-temporal distribution

except for some coastal regions mainly located in marginal seas. Our results also revealed a relatively good agreement between $pCO_2$ from MOM6-COBALT, coastal-SOM-FFN and Socatv6, both in time and space, and most of the discrepancies between the three products are found in regions with poor data coverage, such as in the Bay of Bengal, The Sea of Okhotsk or in the Hudson Bay (Fig. 1a). This study thus provides an objective framework to identify regions where new observational data collections are currently most needed to improve our global understanding of the $CO_2$ exchange between coastal regions and

the atmosphere. From the model-data evaluation, we identified regions where the MOM6-COBALT model shows highest skill in reproducing the spatial and seasonal $pCO_2$ variability, and where the different processes governing the $pCO_2$ dynamics can be examined with reasonable confidence ('high' and 'medium' skill regions in Table 1 and Fig. 1a).



We also adapted a novel method to quantify the contributions of the different physical and biological processes governing the
sea surface $pCO_2$ seasonality in the coastal domain. This method goes one step further than past coastal studies (e.g., Signorini
et al., 2013; Turi et al., 2014) where the processes attribution was only based on the seasonal changes in DIC, ALK, SST and
SSS or/and combined with a series of sequential simulations isolating one term after the other. In particular, our simulations
are non-sequential and allow accounting for the co-variations between the different variables impacted by each process and
how their simultaneous evolution modulates in quantitative terms the $pCO_2$ dynamics. Our approach, which is illustrated in
three coastal regions (the East coast of the U.S, the California Current and the Norwegian Basin), allows to decipher the
complex interplay between ocean transport of chemical species (DIC, ALK and SSS), biological drawdown, fresh water fluxes
(dilution/concentration effects) and thermal changes (air-sea fluxes and transport of temperature) on the $pCO_2$ dynamics.
Depending on the season and region, these terms can reinforce or oppose each other and act to strengthen or dampen the
amplitude of $pCO_2$ seasonal variations that control the air-sea $CO_2$ exchange. Along the East Coast of the U.S and in the
Californian Current, $pCO_2$ increases from winter-to-summer. In the former region, this increase is controlled by a subtle
balance between biological drawdown, thermal changes and ocean circulation, while in the Californian Current, the circulation
due to the upwelling (supplying $pCO_2$-rich waters to the surface) drives the increase in $pCO_2$. In contrast, in the Norwegian
Basin, biological drawdown dominates the marked spring $pCO_2$ decrease observed in the region. These differences in the
quantitative controls of $pCO_2$ dynamics from one region to another support our proposed analysis at the broad scale of the 45
MARCATS regions that together compose the global coastal ocean.

A handful of observational-based studies analyzed the seasonal variability of $pCO_2$ in the global coastal ocean (Cao et al.,
2020; Chen and Hu, 2019; Laruelle et al., 2017). The mechanistic understanding of seasonal $pCO_2$ variations was, and remains
limited by the amount of available observations. The modeling approach tailored for the coastal ocean presented in this
manuscript complements observational studies and help improve our quantitative understanding of the underlying physical
and biological drivers of the coastal $pCO_2$ dynamics. The comparison of the model performance to a state-of-the-art coastal
$pCO_2$ database and continuous $pCO_2$ data product also lends confidence in our model results for a large fraction of the global
coastal domain. The coastal ocean is under tremendous anthropogenic pressure (e.g. climate, land-use change and agriculture,
pollution, urbanization; e.g., Mackenzie et al., 2005; Regnier et al., 2013; Seitzinger et al., 2005). Understanding the interplay
between physics, biology and thermal processes and how they control coastal $pCO_2$ worldwide will be key to assess how their
future changes impact air-sea $CO_2$ exchange in coastal environments.

**Acknowledgements**

L. Resplandy and E. Liao acknowledge the Cooperative Institute for Modeling the Earth System between NOAA GFDL and
Princeton University, the Sloan Research foundation and the Princeton Catalysis Initiative. G. G. Laruelle is research associate
of the F.R.S-FNRS at the Université Libre de Bruxelles. P. Regnier received financial support from BELSPO through the





project ReCAP, which is part of the Belgian research program FedTwin and from the European Union's Horizon 2020 research and innovation program VERIFY (grant agreement no. 776810) and ESM2025 projects

**Data availability**

The Surface Ocean $CO_2$ Atlas (SOCAT) is an international effort, endorsed by the International Ocean Carbon Coordination Project (IOCCP), the Surface Ocean Lower Atmosphere Study (SOLAS) and the Integrated Marine Biosphere Research (IMBeR) program, to deliver a uniformly quality-controlled surface ocean $CO_2$ database. The many researchers and funding agencies responsible for the collection of data and quality control are thanked for their contributions to SOCAT. Every previous version of the SOCAT database can also be accessed from the following page: https://www.socat.info/index.php/previous-

versions/. The coastal-SOM-FFN $pCO_2$ datasets description and dataset can be downloaded from Laruelle et al. (2017). The SST and SSS used for the evaluation the model were extracted from the NOAA OI SST V2 (Reynolds et al., 2007) and the EN4 SSS (Good et al., 2013), respectively. Nutrients data were extracted from the World Ocean Atlas 2018 (Garcia et al., 2019). The delineation and description of the MARCATS segmentation can be found in Laruelle et al. (2013).

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





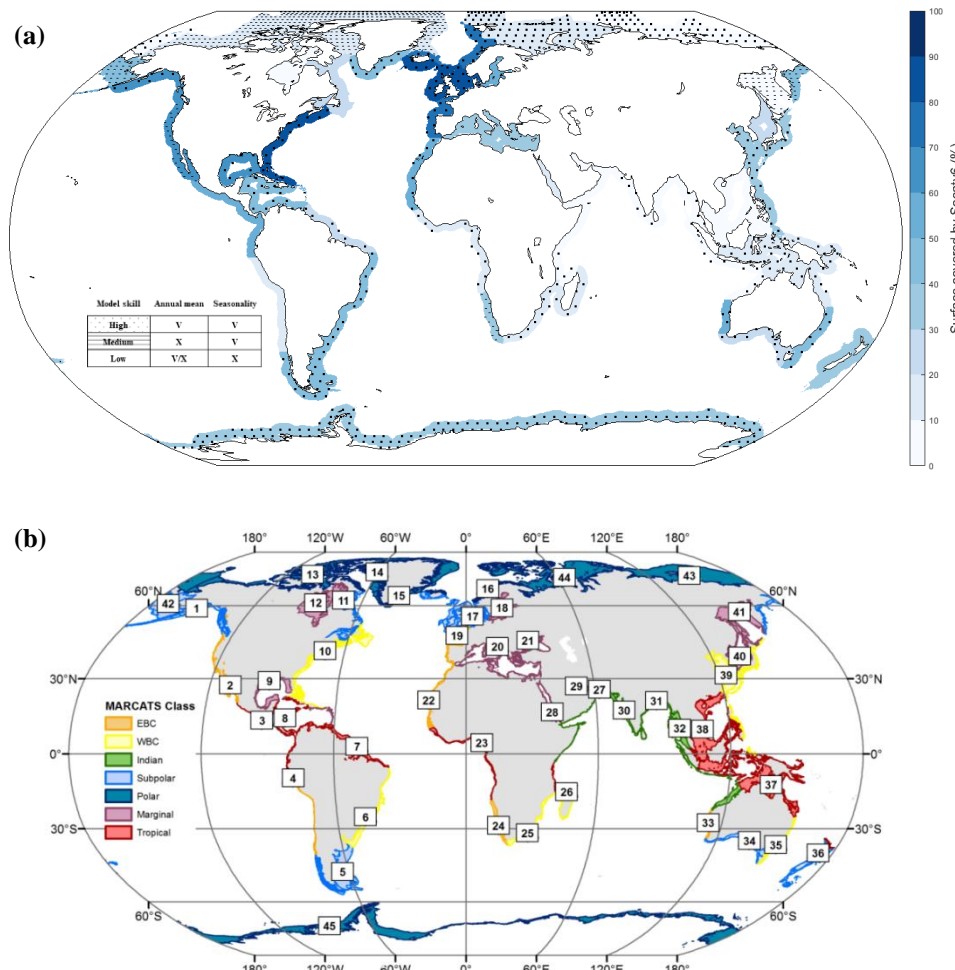

Figure 1: (a) Data coverage (color) and model skill (symbols) in coastal MARCATS (Margins and CATchment Segmentation) regions. The blue intensity indicates the fraction of the MARCATS' surface area covered by Socatv6 observations (from light to dark blue). Dots indicate where the model fulfils three evaluation criteria ('high' skill regions) on the spatio-temporal $pCO_2$ distribution (i.e., annual mean mismatch < 20 µatm between MOM6-COBALT and coastal-SOM-FFN, Pearson correlation coefficient > 0.5 and seasonal amplitude mismatch < 20 µatm). Dashes indicate where the model only fulfils two criteria (seasonal amplitude and phase, 'medium' skill). Other's regions ('low' skill with no symbol) do not fulfil the two criteria associated with seasonality. Details on model skill are in Table 1. (b) Discretization of the coastal seas into 45 MARCATS (Laruelle et al., 2013) grouped into seven classes: Eastern (MARCATS 2, 4, 19, 22, 24, and 33) and Western (MARCATS 6, 10, 25, 35, and 39) boundary currents (EBC and WBC, respectively), polar (MARCATS 13, 14, 15, 16, 43, 44, and 45) and subpolar margins (MARCATS 1, 5, 11, 17, 34, 36, and 42), tropical margins (MARCATS 3, 7, 8, 23, 26, 37, and 38), Indian margins (MARCATS 27, 30, 31, and 32), and marginal seas (MARCATS 9, 12, 18, 20, 21, 28, 29, 40, and 41).





**Figure 2: Observed (center) and modeled (left) spatial distributions of the annual mean state of SST (°C), SSS (no unit), nitrate (NO₃, μmol kg⁻¹), phosphate (PO₄, μmol kg⁻¹) and silicate (SiO₄, μmol kg⁻¹), and model annual mean bias (right). Observational SST and SSS fields are from the NOAA OI SST V2 (Reynolds et al., 2007) and the EN4 SSS (Good et al., 2013). Observational nutrients are from the World Ocean Atlas version 2018 (Garcia et al., 2019). The bias is the difference between MOM6-COBALT and observed values (red indicate regions where the simulated variables by MOM6-COBALT exceed observed values).**

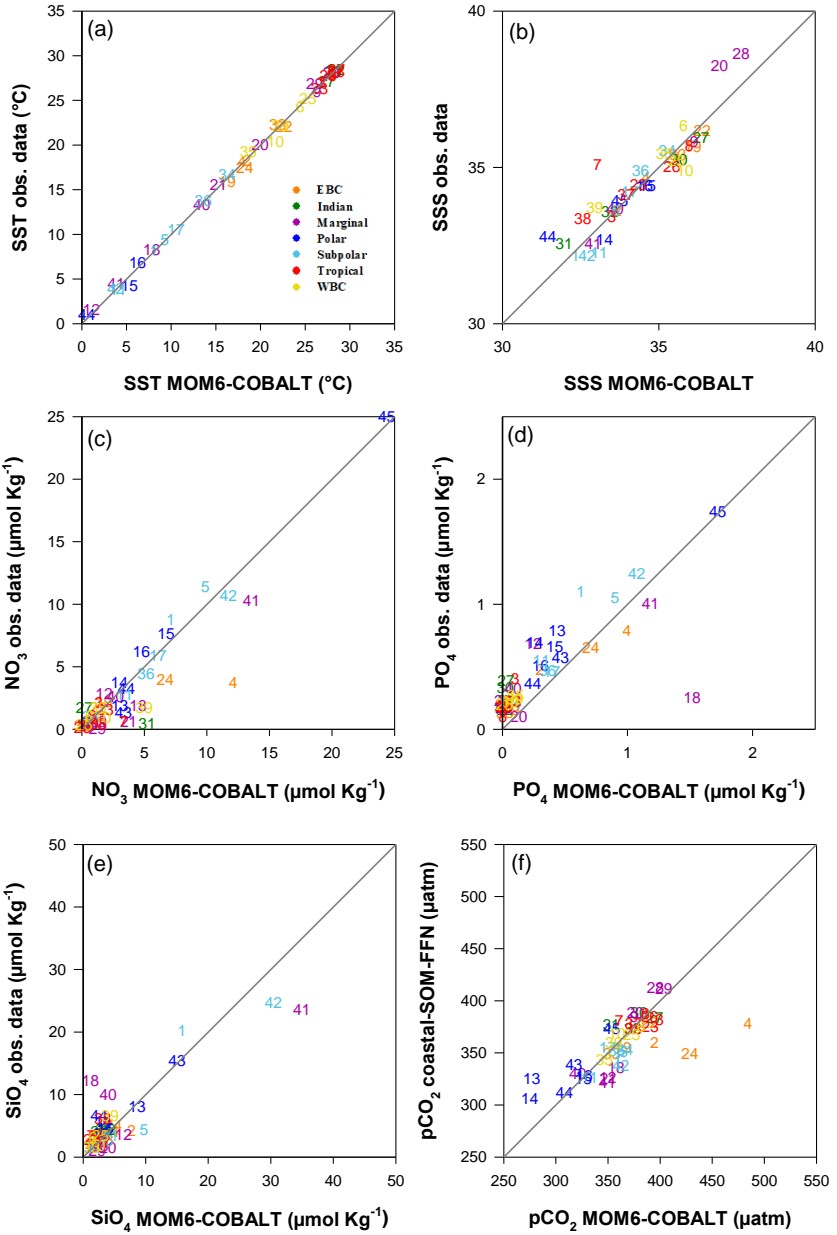

**Figure 3: Comparison between observed and simulated annual mean fields in the 45 MARCATS regions: (a) SST (°C), (b) SSS (no unit), (c) NO₃ (μmol kg⁻¹), (d) PO₄ (μmol kg⁻¹), (e) SiO₄ (μmol kg⁻¹) and (f) pCO₂ (μatm). Observational datasets: SST and SSS are from the NOAA OI SST V2 (Reynolds et al., 2007) and the EN4 SSS (Good et al., 2013), nutrients are from the World Ocean Atlas 2018 (Garcia et al., 2019), pCO₂ is from the coastal-SOM-FFN product (Laruelle et al., 2017). Colors correspond to the seven major MARCATS classes (see Fig. 1b). In panels (d) and (e), the Black Sea (M21) is not represented and has a xy coordinated of (0.2; 3.5 μmol kg⁻¹) in panel (d) and (10.3; 83.1 μmol kg⁻¹) in panel (e). The Antarctic shelf (M45) is also not represented in panel (e) (55.0;49.1 μmol kg⁻¹).**








**Figure 4: Spatial distributions of the annual mean pCO₂ (µatm) generated by (a) MOM6-COBALT, (b) extracted from the Socatv6 database and (c) from the coastal-SOM-FFN product (Laruelle et al., 2017). (d) Model bias as difference between panels (a) and (c) in µatm (red colors correspond to regions in which the pCO₂ simulated by MOM6-COBALT is higher than coastal-SOM-FFN).**






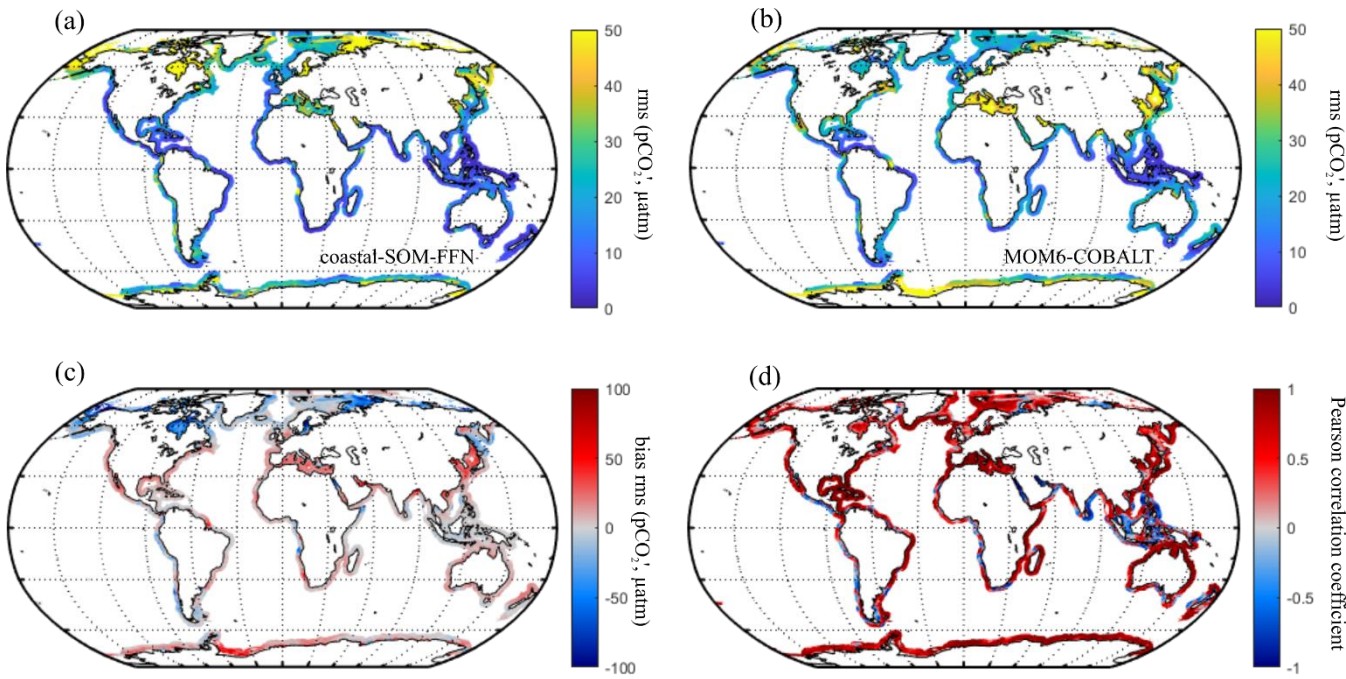

**Figure 5: Seasonal variability in ocean pCO$_2$ (µatm). Seasonal amplitude (a) in the coastal-SOM-FFN product, (b) simulated by MOM6-COBALT model, (c) bias between model and coastal-SOM-FFN seasonal amplitude (red indicate simulated amplitude**

**exceeds coastal-SOM-FFN). The seasonal amplitude is expressed as the root-mean-square of the monthly climatology pCO$_2$ anomalies ($RMS_{pCO_2'}$, µatm). (d) Pearson correlation coefficient of the regression between the seasonal pCO$_2$ cycles calculated by MOM6-COBALT and coastal-SOM-FFN. A value of 1 indicates that both signals are perfectly in phase with one another while a value of -1 represents a complete phase shift.**






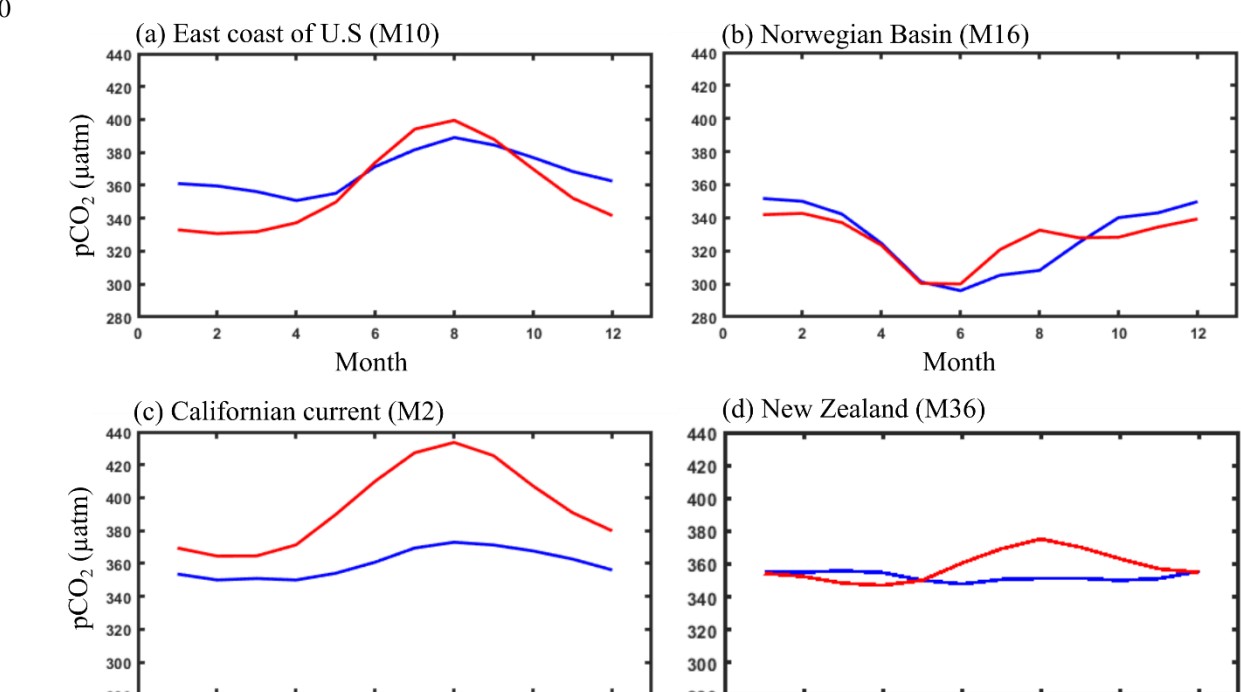

**Figure 6: Seasonal pCO₂ cycle (μatm) derived from coastal-SOM-FFN (in blue) and simulated by MOM6-COBALT (in red) for (a) the East coast of the U.S (M10), (b) the Norwegian Basin (M16), (c) the West coast of North America (M2) and for (d) New Zealand (M36). Month 1 corresponds to January.**






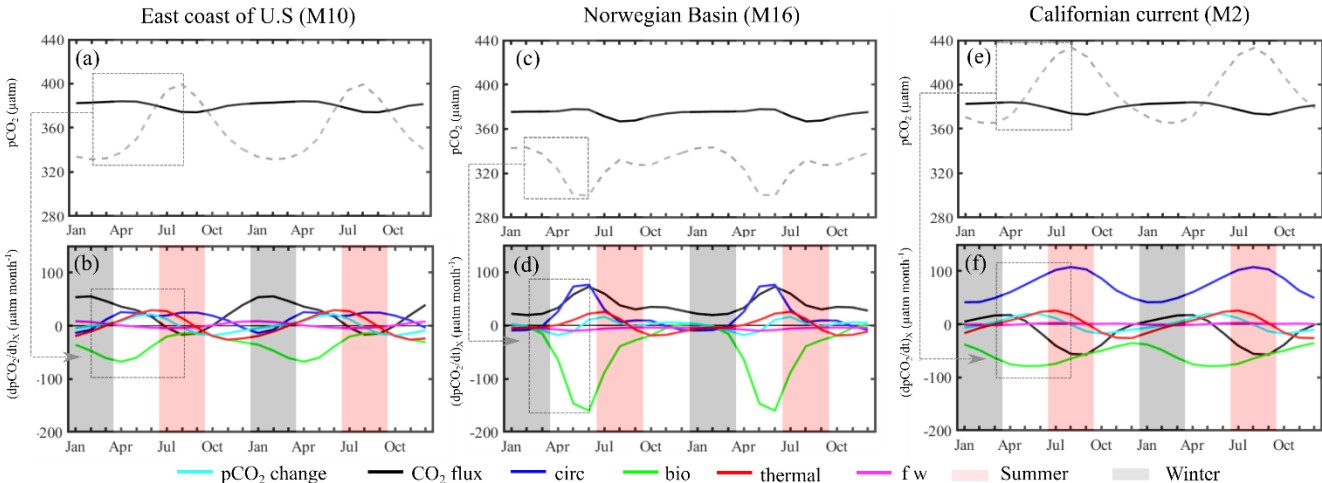

**Figure 7: Processes controlling ocean pCO₂ seasonal cycle.** Mean seasonal sea surface pCO₂ (dashed line) and atmospheric pCO₂ (black line) in µatm simulated by MOM6-COBALT and detrended over (a) the East coast of the US (M10) and (c) the Norwegian sea (M16) and (e) the Californian current (M2). Spatially averaged contributions (in µatm month⁻¹) from biological activity (*bio*, green), temperature changes (*thermal*, red), transport of chemical species (*circ*, blue), freshwater flux (*fw*, pink) and the CO₂ air-sea flux (*CO₂ flux*, black) controlling the pCO₂ temporal change (*pCO₂ change*, cyan) for the three regions (b, d and f). A positive value corresponds to an increase in sea surface pCO₂. Winter corresponds to the months of January, February and March, and Summer to the months of July, August and September.






**Table 1: Model skill level.** For each MARCAT, the model skill ('high', 'medium' and 'low') is attributed from the pCO₂ spatio-temporal analysis. Regions where the model fulfils criteria on the annual mean and seasonality are labelled as 'high' skill regions (i.e., annual mean mismatch < 20 µatm between MOM6-COBALT and coastal-SOM-FFN, Pearson correlation coefficient > 0.5 and seasonal amplitude mismatch < 20 µatm, dots in Fig. 1a). High* skill region can present a bias > 20 µatm on the comparison with Socatv6 (see Table S1). 'Medium' skill regions represent MARCATS where the model only fulfils seasonal criteria (seasonal amplitude and phase, dashed in Fig. 1a). Other's regions ('low' skill) do not fulfil the two criteria associated to the seasonality (no symbol in Fig. 1a). Regions with 'high' model skill are considered as the most robust for an in-depth analysis of the processes driving the coastal pCO₂ dynamics and are highlighted in bold on the Table.

| MARCATS number (Mx) | MARCATS name | MARCATS category | Annual mean pCO₂ (µatm) | | Seasonal pCO₂ | | | Model skill |
|---|---|---|---|---|---|---|---|---|
| | | | | | Amplitude (µatm) | | Phasing (Pearson coefficient) | |
| | | | Coastal-SOM-FFN | Model bias | Coastal-SOM-FFN RMS | Model bias | | |
| 2 | Californian Current | EBC | 360.0 | 34.5 | 8.3 | 16.2 | 1.0 | Medium |
| 4 | Peruvian upwelling Current | EBC | 377.6 | 106.4 | 4.1 | 6.6 | -0.4 | Low |
| **19** | **Iberian upwelling** | **EBC** | **354.8** | **9.3** | **7.5** | **15.6** | **0.8** | **High** |
| **22** | **Moroccan upwelling** | **EBC** | **379.4** | **10.2** | **7.4** | **8.7** | **0.9** | **High** |
| 24 | SW Africa | EBC | 349.1 | 79.3 | 7.2 | 4.2 | 0.9 | Medium |
| **33** | **Leeuwin Current** | **EBC** | **349.4** | **4.2** | **5.6** | **12.7** | **0.9** | **High** |
| 27 | W Arabian Sea | Indian margins | 383.5 | 11.6 | 8.7 | 3.6 | 0.3 | Low |
| **30** | **E Arabian Sea** | **Indian margins** | **388.4** | **-8.3** | **4.8** | **6.2** | **0.7** | **High** |
| 31 | Bay of Bengal | Indian margins | 377.3 | -24.1 | 7.4 | 13.5 | -0.2 | Low |
| **32** | **Tropical E Indian** | **Indian margins** | **373.3** | **0.3** | **2.3** | **5.4** | **0.9** | **High** |
| **9** | **Gulf of Mexico** | **Marginal sea** | **384.3** | **-9.1** | **13.9** | **12.9** | **1.0** | **High** |
| 12 | Hudson Bay | Marginal sea | 326.4 | 5.7 | 65.3 | -46.4 | 0.4 | Low |
| 18 | Baltic Sea | Marginal sea | 336.2 | 21.4 | 79.4 | -44.4 | 0.9 | Low |
| 20 | Mediterranean Sea | Marginal sea | 388.1 | -11.9 | 25.1 | 20.6 | 1.0 | Low |
| 21 | Black Sea | Marginal sea | 325.0 | 25.2 | 141.9 | -116.9 | -0.5 | Low |
| 28 | Red Sea | Marginal sea | 412.2 | -16.5 | 25.0 | -0.4 | -0.9 | Low |
| 29 | Persian Gulf | Marginal sea | 411.2 | -7.6 | 31.3 | 30.7 | -0.9 | Low |
| 40 | Sea of Japan | Marginal sea | 330.3 | -9.3 | 21.1 | 28.0 | 0.9 | Low |
| 41 | Sea of Okhotsk | Marginal sea | 321.2 | 29.2 | 28.6 | -6.5 | 0.7 | Medium |
| 13 | Canadian Archipelago | Polar | 325.4 | -53.1 | 43.4 | -18.0 | 0.9 | Medium |





| 14 | N Greenland | Polar | 306.0 | -24.3 | 21.7 | -9.0 | 0.8 | Medium |
|----|-------------|-------|-------|-------|------|------|-----|--------|
| **15** | **S Greenland** | **Polar** | **325.2** | **1.3** | **24.5** | **-8.5** | **1.0** | **High** |
| **16** | **Norwegian Basin** | **Polar** | **328.1** | **-0.7** | **19.9** | **-6.1** | **0.9** | **High** |
| **43** | **Siberian Shelves** | **Polar** | **338.2** | **-19.7** | **57.4** | **-15.7** | **0.9** | **High*** |
| **44** | **Barents and Kara seas** | **Polar** | **311.6** | **-3.3** | **24.9** | **-7.4** | **0.7** | **High** |
| **45** | **Antarctic Shelves** | **Polar** | **373.7** | **-17.6** | **22.6** | **13.3** | **1.0** | **High*** |
| **1** | **N-E Pacific** | **Subpolar** | **342.5** | **16.8** | **15.8** | **-4.5** | **0.8** | **High*** |
| **5** | **Southern America** | **Subpolar** | **351.1** | **14.0** | **12.1** | **-6.4** | **0.8** | **High** |
| 11 | Sea of Labrador | Subpolar | 326.3 | 5.5 | 17.0 | 0.8 | 0.2 | Low |
| **17** | **NE Atlantic** | **Subpolar** | **354.4** | **-4.5** | **14.9** | **-8.2** | **0.6** | **High** |
| **34** | **S Australia** | **Subpolar** | **352.7** | **13.5** | **3.7** | **12.8** | **0.9** | **High** |
| 36 | New Zealand | Subpolar | 352.4 | 6.1 | 2.6 | 6.2 | -0.5 | Low |
| 42 | NW Pacific | Subpolar | 337.7 | 25.2 | 36.5 | -19.2 | 1.0 | Medium |
| 3 | Tropical E Pacific | Tropical | 382.2 | 17.2 | 6.9 | 3.1 | 0.3 | Low |
| **7** | **Tropical W Atlantic** | **Tropical** | **380.3** | **-19.8** | **2.8** | **9.6** | **1.0** | **High** |
| **8** | **Caribbean Sea** | **Tropical** | **387.6** | **-1.7** | **6.6** | **2.2** | **1.0** | **High** |
| **23** | **Tropical E Atlantic** | **Tropical** | **374.6** | **15.9** | **2.9** | **1.5** | **0.6** | **High*** |
| **26** | **Tropical W Indian** | **Tropical** | **384.8** | **4.8** | **7.1** | **5.6** | **0.9** | **High*** |
| **37** | **N Australia** | **Tropical** | **378.5** | **-4.0** | **4.3** | **5.2** | **1.0** | **High** |
| 38 | SE Asia | Tropical | 373.5 | 0.6 | 2.6 | 8.9 | 0.2 | Low |
| **6** | **Brazilian Current** | **WBC** | **374.8** | **7.0** | **6.7** | **7.5** | **0.9** | **High** |
| **10** | **East coast of US** | **WBC** | **368.1** | **-9.6** | **12.0** | **12.4** | **0.9** | **High** |
| **25** | **Agulhas Current** | **WBC** | **367.1** | **5.7** | **7.1** | **8.1** | **1.0** | **High** |
| **35** | **E Australian Current** | **WBC** | **343.9** | **2.9** | **3.3** | **7.4** | **1.0** | **High** |
| **39** | **China Sea and Kuroshio** | **WBC** | **359.6** | **-4.1** | **10.3** | **13.2** | **0.9** | **High** |
