# Peer review of "A framework to evaluate and elucidate the driving mechanisms of coastal sea surface pCO2 seasonality using an ocean general circulation model (MOM6-COBALT)"

_Ocean Science, 2021_

## Author Comment (AC1)

**Authors response: We thank reviewer #1 for his valuable comments and for highlighting the importance of the work presented in this study in the context of the global carbon budget.**

**We addressed all comments raised by the reviewer and provide a point-by-point answer below (in blue). Changes and additions to the original manuscript have been introduced using the Word's "track changes" option and the line numbers noted in our answers refer to the revised manuscript with the track changes option. We also took this opportunity to correct several typos in the manuscript.**

**On behalf of the co-authors,**
**Alizée Roobaert**

**Reviewer #1 Evaluations:**

This paper thoroughly evaluated the model performance in the coastal region. Then, it examined the drivers of pCO2 seasonal variations in several coastal regions using the decomposition method recently proposed.

Recent studies have shown that the CO2 uptake in the coastal ocean cannot be ignored in the global CO2 budget. The detailed analysis the variability of the coastal CO2 flux has been needed.

The manuscript is well organized and easy to follow.

- **R1C1: My concern is just that the decomposition results are shown in the only three coastal regions.**

As the authors mentioned, uncertainty of the reconstructed pCO2 dataset is not small especially in the data limited region. Therefore the model performance is not necessarily doubtful even if the model output is not consistent with the observation-based estimates.

As long as the discrepancy is clearly stated, the decomposition result in other regions and the detailed discussion of the geographical distribution of the driving force is useful for our understanding.

**R1R1: We understand the reviewer's comment regarding the opportunity offered by our new method to analyze the seasonal variability of pCO₂ in others coastal regions than the ones presented in our study. The primary objective of this study is to introduce the coastal-tailored approach to quantify biological and physical contribution to pCO₂ changes and demonstrate that it works by showcasing a few case studies where the model performance is good compared to available observations and existing literature.**

**Naturally, we agree with the reviewer that the performance of the model is not necessarily doubtful in region with poor coverage in the pCO₂-based dataset and that investigating the rest of the global coastal ocean is a worthy endeavor. This first manuscript presents and evaluate the new methodology in detail, in line with the scope of the Ocean Sciences journal, and will allow to investigate coastal seasonal pCO₂ variability at the global scale in a second step by referring to this paper. We have, however, significantly extended the model to data comparison by adding direct comparison to raw SOCATv6 data in MARCATS that are well sampled, but also at 4 coastal sites (Antarctic Peninsula, Queensland Plateau NE Australia, Papua New Guinea and Terra Nova) where SOCATv6 data have a good spatio-temporal coverage even if they are located in MARCATS that are poorly sampled (See our new Fig. 6). Finally, we have changed the wording "model skill" to "model to coastal-SOM-FFN agreement" everywhere in the main text, figures**

**and tables, to emphasize that a poor agreement does not equate a poor skill and added the following sentence in line 387-390:**

"Note that the model-SOCATv6 seasonal evaluation in Terra Nova presents a good agreement although the MARCATS scale (Sea of Labrador, M11) evaluation to which this region belongs to reveals a low agreement, showing that a poor agreement between coastal-SOM-FFN and the model does not equate to poor model skill when these regions are under sampled by SOCATv6."

Other minor comments are follows;

- **R1C2: Line 139 and many others, "Socatv6": "SOCATv6" would be better.**

**R1R2: We agree with R1C2, and this has been changed accordingly everywhere in the main text, figures and in the supplementary material.**

- **R1C3: Figure 1a: Dots and dashes in the inserted table are not similar with those in the main body of Figure 1**

**R1R3: Indeed, there is an inconsistency in Fig. 1a between the dots patterns used on the map at the top of the figure to represent the level of 'model to coastal-SOM-FFN agreement' and the table presenting the legend. We thank the reviewer for pointing this out and we have updated the figure accordingly (see below).**

**Updated figure:**

[Figure]

**Figure 1: (a) SOCATv6 spatial  coverage (color) and agreement between model and coastal-SOM-FFN product (symbols) in coastal MARCATS (Margins and CATchment Segmentation) regions. The blue intensity indicates the fraction of the MARCATS' surface area covered by SOCATv6 observations (from light to dark blue). Dots indicate where the model fulfils three evaluation criteria ('high'  agreement regions) on the spatio-temporal $pCO_2$ distribution (i.e., annual mean mismatch < 20 µatm between MOM6-COBALT and coastal-SOM-FFN, Pearson correlation coefficient > 0.5 and seasonal amplitude mismatch < 20 µatm). Dashes indicate where the model only fulfils two criteria (seasonal amplitude and phase, 'medium' agreement). Other's regions ('low'  agreement with no symbol) do not fulfil the two criteria associated with seasonality. Details on model to coastal-SOM-FFN agreement are in Table 1. (b) Discretization of the coastal seas into 45 MARCATS (Laruelle et al., 2013) grouped into seven classes: Eastern (MARCATS 2, 4, 19, 22, 24, and 33) and Western (MARCATS 6, 10, 25, 35, and 39) boundary currents (EBC and WBC, respectively), polar (MARCATS 13, 14, 15, 16, 43, 44, and 45) and subpolar margins (MARCATS 1, 5, 11, 17, 34, 36, and 42), tropical margins (MARCATS 3, 7, 8, 23, 26, 37, and 38), Indian margins (MARCATS 27, 30, 31, and 32), and marginal seas (MARCATS 9, 12, 18, 20, 21, 28, 29, 40, and 41).**

---

## Author Comment (AC2)

**Authors response: We thank reviewer #2 for his valuable comments and for highlighting the importance of the work presented in this study in the context of the global carbon budget.**

**We addressed all comments raised by the reviewer and provide a point-by-point answer below (in blue). Changes and additions to the original manuscript have been introduced using the Word's "track changes" option and the line numbers noted in our answers refer to the revised manuscript with the track change option. We also took this opportunity to correct several typos in the manuscript.**

**On behalf of the co-authors,**
**Alizée Roobaert**

**Reviewer #2 Evaluations:**

Roobaert et al. assess the skill of the MOM6-COBALT model for representing the seasonal cycle of pCO2 in coastal regions and develop a methodology for interrogating the processes driving seasonal and regional differences. They use the model output to interrogate the drivers of the seasonal signal in three regions where model skill is high. This study makes good use of data products for assessing the skill of models in reproducing seasonal coastal dynamics and the unique information that models can bring to coastal carbon research, however, a few issues should be addressed before publication.

- **R2C1: 1) In the method to assess the different processes controlling seasonal pCO2 variability, the assumption that the coefficients (explained in lines 195-206) are constant in time needs to be explained.**

Are the coefficients truly constant in time if the goal is to understand how processes like freshwater discharge (a spring event in many regions) impact pCO2? Doesn't a spring-time river runoff event, for example, change the relationship between DIC, ALK, SSS, etc, which would not be reflected in coefficients derived from average conditions over 1998-2015?

**R2R1: We agree with the reviewer that residual biases between our regression-based reconstruction and the model pCO₂ can be attributed to strong seasonal signal in river discharge. The coefficients we use are indeed constant in time. We evaluated the impact of using coefficients that are both time and space varying and found that it further reduces biases tied to river discharge in the Amazon plume for instance. This improvement is, however, marginal compared to using the simpler approach we chose here (space-varying only). These residual biases are probably caused by the non-linearities associated with the large seasonal changes that the Takahashi linearization approach cannot capture even with time-varying coefficients. We agree with the reviewer that this is however an important point to make in the manuscript. We therefore added a panel to Figure S1 that shows the pCO₂ in the model and the pCO₂ reconstructed using the three different methods (traditional space varying coefficients from Sarmiento and Gruber, space varying regression method used here, and space and time varying regression method) at a point in the Amazon plume and added a paragraph to the method section that discusses this point which now reads as follows (lines 200-230):**

[revised manuscript text omitted]

the regression-based approach developed in this study to compute space varying sensitivity coefficients (using the CO2SYS program, see section 2.3 for details). (d) The difference in bias between the traditional and regression-based approaches shows a strong reduction in biases when using the regression-based method. Biases (in µatm) are quantified using the root mean square error (RMSE) between the $pCO_2$ simulated by the model and the $pCO_2$ reconstructed from simulated monthly SST, DIC, ALK and SSS (Eq. 2). (e) time series of seasonal $pCO_2$ anomaly at 310.25°E, 1°N (star on panel d) simulated by the MOM6 model (black), and reconstructed using the space varying coefficients of Sarmiento and Gruber 2006 (blue), using the space varying regression-based coefficients used in this study (red), and space and time varying regression-based coefficients (purple). See text in method section for further details.

- **R2C2: 2)** In the methodological limitations section (3.1.4) it is mentioned the coastal-SOM-FFN climatology does have limitations in reproducing pCO2 variability in some regions. Since this is the case, for the regions where there are SOCATv6 data (lines 325-326 state there are 45 grids with sufficient data), **the paper should include model-SOCATv6 comparisons, especially for seasonal amplitude.**

Right now, Figure 4 does show SOCATv6 annual mean but Figure 5 does not show seasonal amplitude from SOCATv6, and seasonal amplitude is, as the authors state, underrepresented by coastal-SOM-FFN. Figures 4 and 5 should both include SOCATv6 as well as the residuals between model and SOCATv6 (for the regions where there is observational data). This is also an issue with the supplemental tables, where Table S1 presenting annual mean does include SOCATv6 but S2 presenting seasonal amplitude does not. The paper should include a more robust assessment of model-SOCATv6 seasonal amplitude comparisons, given seasonality, not annual mean, is the central focus of the study.

**R2R2: In this comment, the reviewer highlights that the $pCO_2$-based product is limited in poorly sampled regions and that a more direct comparison of the $pCO_2$ seasonality simulated by the model against the SOCATv6 database would be valuable. This idea is also echoed in several later comments from the reviewer regarding a suggestion to update Fig. 4 (R2C5), the addition of a seasonal evaluation based on 45 grid cells with continuous SOCATv6 data (R2C6) or the addition of seasonal $pCO_2$ cycles from SOCATv6 in Fig. 6 (R2R12).**

**We looked at $pCO_2$ measurements extracted from time-series at mooring stations that could provide validation datasets for our model. A recent literature review (Sutton et al., 2019) provides an extensive overview of these stations, but as illustrated on this map extracted from the NOAA website (https://www.ncei.noaa.gov/access/ocean-carbon-data-system/oceans/Moorings/ndp097.html) the vast majority of these stations fall outside of the geographic limits of our coastal domain and data collection at mooring stations in the coastal domain, such as along the north American coast only began, at the earliest (2004), mid-way through the time period represented by our study (1998-2015).**

[Figure]

**Figure 1.** (a) Locations of moored autonomous seawater $pCO_2$ and pH time series included in Sutton et al. 2018. (b) U.S. West Coast and (c) Hawaii locations. Blue symbols are open ocean sites; green are coastal shelf; and orange are coral reefs.

Figure from **https://www.ncei.noaa.gov/access/ocean-carbon-data-system/oceans/Moorings/ndp097.html**

Nevertheless, following the reviewer's suggestion we addressed the reviewer's concern by expanding our validation of the results using raw SOCATv6 data.

1) First, we updated our Fig. 6 and now present at the spatial resolution of 0.25 degree, the number of climatological months where at least one $pCO_2$ measurement is available based on the climatological seasonal cycle derived from the SOCATv6 database (new Fig. 6a). We believe this map provides valuable insight to the reader regarding both the spatial and temporal heterogeneity of the data coverage of the SOCATv6 database.

2) At the MARCATS scale, we selected 11 MARCATS with a good spatio-temporal coverage of the SOCATv6 database and for which is it possible to obtain a complete climatological seasonal cycle (see our new Fig. 6). The 11 MARCATS regions are the Californian Current (M2), Tropical E Pacific (M3), the Gulf of Mexico (M9), the East coast of US (M10), S Greenland (M15), Norwegian Basin (M16), NE Atlantic (M17), Iberian upwelling (M19), Moroccan upwelling (M22), China Sea and Kuroshio (M39) and New Zealand (M36). For each of these MARCATS, we calculated climatological seasonal $pCO_2$ cycles derived from the model, SOCATv6 and the coastal-SOM-FFN product (new Figs. 6b-l). For these regions, we calculated the bias between the seasonal amplitude of MOM6-COBALT and the one of SOCATv6 as well as their respective Pearson correlation coefficient. All these new values have been added to the updated Table S2 (see at the end of this document) in bracket to provide the reader with additional information regarding the model performance wherever there were enough field data to evaluate the model against Socatv6. Note that we did not perform the SOCATv6-model seasonal evaluation (bias and Pearson correlation coefficient) for the others MARCATS because of the lack of temporal coverage (new Fig. 6a). This is also the reason why we did not update Fig. 5 with a map of the seasonal $pCO_2$ amplitude from SOCATv6 as proposed by R2.

3) Finally, we selected 4 sites that are that are significantly smaller than the MARCATS scale and located in regions where the seasonal evaluation against SOCATv6 is not possible at the MARCATS scale. These sites are located off the Antarctic Peninsula, on the Queensland Plateau in NE Australia, in coastal waters of Papua New Guinea and of Terra Nova (see black boxes in new Fig. 6a). These sites present a complete seasonal climatological $pCO_2$ signal derived from SOCATv6 which can be compared to the model. The seasonal cycles for these four sites have been added to the updated Fig. 6 (panels m-p).

**Updated Fig. 6:**

[Figure]

Figure 6: (a) SOCATv6 temporal coverage evaluated as the number of months (1 to 12) where at least one pCO$_2$ measurements is available (see details in methods). Seasonal pCO$_2$ cycle (µatm) derived from SOCATv6 (bar in grey), coastal-SOM-FFN (in blue),  and simulated by MOM6-COBALT (in red) for several MARCATS (b-l) and four coastal sites of smaller spatial extent than MARCATS (m-p). The location of the four coastal sites is represented in black boxes in panel (a).  Month 1 corresponds to January. For consistency of y axis between panels, the value of 276 µatm is not represented in panel (p) for month 5 for the SOCATv6 data.

**Overall, as a consequence of this new evaluation strategy, Fig. 4 (see R5C5), Fig. 6, and Table S2 have been updated as well as the following sections of the manuscript:**

**Lines 164-170:**

"We use two metrics to evaluate SOCATv6 spatial and temporal coverage. First, we evaluate the spatial coverage at the MARCATS scale by computing the percent surface area sampled by SOCATv6 data for each MARCATS. A 50 % spatial coverage means that SOCATv6 data are available in 50 % of the 0.25° x 0.25° cells included in this specific MARCATS (this metric is used in Fig. 1a). Second, we evaluate the ability of SOCATv6 to capture the seasonality at the grid cell scale by computing the number of months where at least one SOCATv6 pCO$_2$ measurement for each 0.25° x 0.25° grid cells. A 8-months temporal coverage means that 8 out of the 12 months are sampled at least once in this grid cell (this metric is used in Fig. 6a)."

**lines 349-392:**

"Our analysis reveals that the seasonal amplitudes simulated by MOM6-COBALT are systematically larger than the ones estimated by the coastal-SOM-FFN product (Fig. 5a-b, red colors in Fig. 5c and positive biases in Table S2) for all coastal regions belonging to EBC, WBC, Indian and tropical margins. For the majority of polar and subpolar margins and for some marginal seas, the model simulates lower seasonal $pCO_2$ amplitudes (blue colors in Fig. 5c and negative biases in Table S2).  Quantitatively, absolute biases between the modelled and coastal-SOM-FFN amplitudes do not exceed 20 µatm except for marginal seas where larger discrepancies are calculated (6 of the 9 marginal MARCATS, Table S2). The monthly mean $pCO_2$ seasonal cycle simulated by MOM6-COBALT is also well in phase (Pearson correlation coefficients > 0.5) with the one extracted from coastal-SOM-FFN in 34 out of the 45 MARCATS (red colors in Fig. 5d and Table S2). The agreement is especially good in the best monitored MARCATS regions (MARCATS where > 50 % of the area is covered by SOCATv6 observations, Table S1). For instance, in regions with good data coverage such as along the East coast of the U.S (M10), the Norwegian Basin (M16), the Californian Current (M2), the Leeuwin Current (M33), or the Brazilian Current (M6), the Pearson correlation coefficient is higher than 0.9 (Table S2). In contrast, the seasonal $pCO_2$ cycle simulated by MOM6-COBALT substantially diverges from that of the coastal-SOM-FFN in four poorly monitored marginal seas and in a few of the EBCs, Indian margins, subpolar margins, and tropical margins (Pearson correlation coefficient < 0.5, Table S2 and blue colors in Fig. 5d).

The model $pCO_2$ seasonal evaluation against SOCATv6 is only performed in 11 MARCATS namely the Californian Current (M2), Tropical E Pacific (M3), the Gulf of Mexico (M9), the East coast of US (M10), S Greenland (M15), Norwegian Basin (M16), NE Atlantic (M17), Iberian Upwelling (M19), Moroccan upwelling (M22), China Sea and Kuroshio (M39) and New Zealand (M36). The modeled seasonal cycle is in good agreement with that one derived from SOCATv6 (Fig. 6b-n, Table S2) with absolute biases < 20 µatm for all of the 11 selected MARCATS and Pearson correlation coefficients close to 0.5 or higher except for the Iberian Upwelling (M19, Pearson value of 0.2) and in the New Zealand shelf (M36, value of 0.3). We did not perform the SOCATv6-model seasonal evaluation for the other MARCATS because the vast majority of grid cells only include data for less than 4 climatological months (Fig. 6a). However, ~~agreement is especially good in the best monitored MARCATS regions (MARCATS where > 50 % of the area is covered by Socatv6 observations, Table A1). For instance, in regions with good data coverage such as along the East coast of the U.S (M10, Fig. 6a), the Norwegian Basin (M16, Fig. 6b), the Californian Current (M2, Fig. 6c), the Leeuwin Current (M33), or the Brazilian Current (M6), the Pearson correlation coefficient is higher than 0.9 (Table A2). In contrast, the seasonal $pCO_2$ cycle simulated by MOM6-COBALT substantially diverges from that of the coastal-SOM-FFN in four poorly monitored marginal seas (M12, M21, M28, M29)-~~ sites of smaller spatial extent than MARCATS for which we calculated climatological seasonal $pCO_2$ signals from the SOCATv6 dataset and compared them with the model $pCO_2$. These sites are located off the Antarctic Peninsula, on the Queensland Plateau in NE Australia, in coastal waters of Papua New Guinea and of Terra Nova (see black boxes in Fig. 6a). In those regions, the absolute biases on the seasonal amplitude between MOM6-COBALT and SOCATv6 (Figs. 6m-p) are less than 20 µatm and the phase in the seasonal cycles present a good agreement with a Person correlation coefficient value of 0.8 except for the Papua New Guinea (value of 0.5). Note that the model-SOCATv6 seasonal evaluation in Terra Nova presents a good agreement although the MARCATS scale (Sea of Labrador, M11) evaluation to which this region belongs to reveals a low agreement, showing that a poor agreement between coastal-SOM-FFN and the model does not equate to poor model skill when these regions are under sampled by SOCATv6.

"

**Lines 446-451:**

"This lack of observations could partly explain why MOM6-COBALT-coastal-SOM-FFN $pCO_2$ biases exceed 20 µatm in these regions. The seasonal model evaluation against raw SOCATv6 is limited at the MARCATS scale and mainly performed against coastal-SOM-FFN due to the very few coastal regions that contain a continuous climatological seasonal $pCO_2$ cycle (Fig. 6a) in the SOCATv6 database. This study highlights the regions (Fig.1a, e.g., Indian ocean margins, Peruvian upwelling, marginal seas) where new observational data are most urgently needed, specifically collected during periods of the years that are currently not covered to improve our understanding of the $CO_2$ exchange between coastal regions and the atmosphere at the regional and global scales."

- **R2C3: 3) The ESRL atmospheric data are not properly cited.**

First, ESRL does not provide pCO2 as stated in line 115. The atmospheric community measures and provides xCO2. The authors need to properly cite the ESRL data source (not the current citation of Joos and Spahni) and explain how atmospheric pCO2 was calculated.

**R2R3: We thank the reviewer for drawing our attention to this issue in the original text. It should read $xCO_2$ instead of $pCO_2$ from ESRL. $xCO_2$ was converted to $pCO_2$ using atmospheric pressure and water vapor pressure by the model. The ESRL $xCO_2$ is from the NOAA Marine Boundary Layer (MBL) (https://gml.noaa.gov/ccgg/mbl/index.html). We modified the description and added the following two references.**

**Conway, T.J., P.P. Tans, L.S. Waterman, K.W. Thoning, D.R. Kitzis, K.A. Masarie, and N. Zhang, 1994, Evidence for interannual variability of the carbon cycle from the NOAA/CMDL global air sampling network, J. Geophys. Res., 99, 22831-22855.**

**GLOBALVIEW-CO2: Cooperative Atmospheric Data Integration Project - Carbon Dioxide. CD-ROM, NOAA ESRL, Boulder, Colorado [Also available on Internet via anonymous FTP to ftp.cmdl.noaa.gov, Path: ccg/co2/GLOBALVIEW], 2011. See version history (gml.noaa.gov/ccgg/globalview/co2/co2_version.html).**

**We modified the text lines 116-120:**

"The ocean model is forced by the 55-km horizontal resolution Japanese atmospheric reanalysis (JRA55-do) version 1.3 at a 3-hour frequency between 1959 and 2018 (Tsujino et al., 2018), and the atmospheric  concentration data (xCO₂) from the Earth System Research Laboratory  (Conway et al., 1994; GLOBALVIEW-CO2, 2004). The xCO₂ is converted to pCO₂ using atmospheric and water vapor pressures by the model."

Minor issues:

- **R2C4: Line 45: This statement seems to be Northern Hemisphere biased.**

Given this study used a SOCAT-based data product, Southern Hemisphere coastal regions are extremely underrepresented and many areas are likely not well characterized.

**R2R4: The study by Roobaert et al. (2019) is indeed indirectly based on the SOCAT data product. However, we would like to stress out that we used the data product of Laruelle et al. (2017) which is a continuous $pCO_2$ climatology in space and time that fills out coastal regions devoid of data and underrepresented in the SOCAT database. This choice was motivated by the need to work with a $pCO_2$ field that included the entirety of the world's coastal ocean and would thus not be skewed by the current spatial heterogeneity of the SOCAT database. In their analysis of the $FCO_2$ seasonality, Roobaert et al. (2019) show a 6-month shift on the seasonal signal between "low" latitudes (40° N - 40° S) and high latitudes (> 60°). The latter latitudinal band has a more intense $CO_2$ sink occurring in summer both in the Northern and Southern hemispheres (see panels (a) and (d) in Fig. 5 of Roobaert et al. 2019). The more intense global coastal sink that occurs in summer that we associated only with the Northern Hemisphere, is in our view not a bias due to an under-representation of the southern hemisphere but only from the differing areal distribution of the coastal regions. Indeed, as shown by panel (c) of Fig. 4 of Roobaert et al., a large majority of the coastal surface lies in the high latitude of the Northern Hemisphere.**

[Figure]

[Figure]

Fig. 5 of Roobaert et al. (2019)                                    Fig. 4 of Roobaert et al. (2019)

**We believe that the phrasing of the following sentence (lines 43-48) in the manuscript might have led to a misunderstanding that we hope this updated version will prevent:**

"This study identified that at the annual timescale, the global coastal ocean acts as an atmospheric $CO_2$ sink (-0.2 ± 0.02 Pg C $yr^{-1}$) with a more intense $CO_2$ uptake occurring in boreal summer because of the disproportionate contribution of high latitude coastal regions in the Northern Hemisphere which cover 25 % of the total coastal area and are characterized by an intense $CO_2$ sink in summer. "

**R2C5: Line 290 / Figure 4: As stated earlier, it would be easier to see the model-data difference if a residual plot was included rather than ask the reader to compare Fig 4a and 4b.**

**R2R5: We agree with the reviewer's comment, which is directly connected to R2C2. We modified Fig. 4 and added a new panel (panel c) which represents the difference between the annual mean $pCO_2$ from MOM6-COBALT (Fig. 4a) and SOCATv6 (Fig. 4b). We also updated the text when we refer to this figure.**

**Updated Fig. 4**

[Figure]

**Figure 4: Spatial distributions of the annual mean $pCO_2$ (µatm) generated by (a) MOM6-COBALT, (b) extracted from the SOCATv6 database (c) model bias as difference between panels (a) and (b) in µatm (red/blue colors correspond to regions in which the $pCO_2$ simulated by MOM6-COBALT is higher/lower than SOCATv6). (d) Spatial distribution of the annual mean $pCO_2$ from the coastal-SOM-FFN product (Laruelle et al., 2017). (e) Model bias as**

**difference between panels (a) and (e).**

- **R2C6: Line 326: Many of these 45 grid cells with continuous pCO2 time series are likely buoy locations. Added to SOCAT in 2015, these continuous time series are an essential feature of SOCAT for seasonal assessments like this study and make a strong case for a more thorough model-data comparison as mentioned previously.**

**R2R6: We agree with the reviewer that a model-SOCATv6 seasonal analysis was missing in our original manuscript. We considered this comment and our new evaluation strategy regarding this issue is described in detail in the R2R2 response.**

- **R2C7: Line 352-353: In some places like here the regions are only stated by their associated numbers, however, it is easier for the reader to understand the results if stated by their name and number as in lines 356-357.**

**R2R7: We agree with R2C7 and modified some lines in the text accordingly.**

**Lines 316-320:** "The regions where the bias exceeds this threshold include two EBC's (the Californian (M2) and the Peruvian upwelling (M4) Currents), two marginal seas (the Seas of Japan, M40, and Okhotsk, M41), and one Polar (the Antarctic shelves, M45), subpolar (NW Pacific, M42) and Tropical East Atlantic (M23) shelf."

**lines 343-346:** "These regions belong mainly to EBCs (3 out of the 6 EBC MARCATS), marginal seas (3 out of the 9 marginal seas MARCATS), the remaining four being either polar (the Canadian Archipelago (M13) and the N Greenland ( M14)), subpolar (NW Pacific, M42) or Indian margins (the Bay of Bengal, M31)."

**lines 408-411:** "Note that these MARCATS but the Siberian (M43) and Antarctic, (M45) shelves, the NE Pacific (M1), the Tropical E Atlantic (M23) and the Tropical W Indian (M26) also present an annual mean pCO₂ bias < 20 µatm in the MOM6-COBALT- SOCATv6 and coastal-SOM-FFN- SOCATv6 comparisons (Table S1)"

**Line 480:** "Interestingly however, some regions reveal significant biases in the major environmental fields but not in the pCO₂ (e.g., Tropical W Atlantic, M7) while in other regions, the reverse is observed (e.g., the Mediterranean (M20) and W Arabian, (M27) Seas and in New Zealand (M36))."

- **R2C8: Lines 388-390: This seems to be an important result of the study that should be included in the Conclusion section."**

**R2R8: We agree with the reviewer's remark and introduced an explicit reference to this results in the updated version of the conclusion section in lines 595-600:**

"This study highlights the regions (Fig.1a, e.g., Indian ocean margins, Peruvian upwelling, marginal seas) where new observational data are most urgently needed, specifically data collected during different periods of the year that are currently missing to improve our understanding of the CO₂ exchange between coastal regions and the atmosphere at the regional and global scales."

- **R2C9: Section 3.2.1: Cai et al. 2020 find that different processes drive variation in pCO2 in different subregions of US East Coast. How do these model-based results compare with their data-based assessment of drivers?**

(See: Cai, W.-J., et al. (2020). Controls on surface water carbonate chemistry along North American ocean margins. Nature Communications, 11(1), 2691. https://doi.org/10.1038/s41467-020-16530-z)

**R2R9: We thank the reviewer for drawing our attention to this recent publication. The study of Cai et al. (2020) investigates the dynamics of the carbonate system along the continental margin of North America, thus including 2 of the MARCATS for which we provide a detailed analysis of the drivers controlling the seasonality of pCO₂ (the California current, M2, and the east coast of the US, M10). However, the work of Cai et al. mostly focusses on spatial variations and comparisons between the different coastal regions surrounding North America. In their study, the main references to the seasonal variability of pCO₂ (see page 9 in particular) focusses on the Atlantic coast of North America using a time-varying box model. In this region, the authors describe seasonal variations of pCO₂ similar to the ones reported in our study (with a pCO₂ increase from spring to summer and a decrease during fall leading to minimum pCO₂ values in winter). The brief description of the factors controlling these variations is also in line with the findings of our study with a thermally driven seasonality that is dampened by biological uptake. In the revised version of our manuscript, we now include 2 references to Cai et al.'s study in the section 3.2.1:**

**Line 515:** "This thermal effect was already identified by Signorini et al. (2013) in their observational study and further confirmed by (Cai et al., (2020)."

**Lines 540-542:** 'The importance of the thermal and circulation effects as well as the presence of a strong biological drawdown are in line with results from past studies (e.g., Laruelle et al. (2015), Shadwick et al. (2010, 2011), and Signorini et al. (2013) and (Cai et al., (2020))."

- **R2C10: Lines 575-583: Description of the xCO2 data source is missing from this section.**

**R2R10: The source of the xCO2 data (i.e. Convey et al., 1994; GLOBALVIEW-CO2 project report) has now been introduced into the manuscript has requested by the reviewer.**

**We also added lines 645-647:** "The coastal-SOM-FFN pCO₂ datasets description and dataset can be downloaded from Laruelle et al. (2017) and the atmospheric CO₂ concentration data (xCO₂) derived from the Earth System Research Laboratory (Conway et al., 1994; GLOBALVIEW-CO2, 2004)."

**R2C11: Figure 3: This is another figure that could benefit from showing a MOM6-COBALT vs SOCATv6 comparison for pCO2.**

**R2R11: As stated in our answer to comment R2C5 which suggests that a map presenting the difference between the annual mean pCO₂ of MOM6-COBALT and that of SOCATv6 would be a valuable addition to Fig. 4, we find it difficult to add a panel comparing pCO₂ derived from SOCATv6 vs that one simulated by the model.**

**Indeed, all the panels in Fig. 3 display a comparison of the absolute values of different variables derived from observations versus those simulated by the model. This comparison is carried out at the MARCATS scale using continuous climatologies in space, i.e. each MARCATS is fully covered both in terms of observational data and by the model. Adding a panel comparing the absolute pCO₂ between SOCATv6 and MOM6-COBALT does not seem adequate given that the SOCAT data are spatially discontinuous. In addition, in several MARCATS, no data exist in the SOCATv6 database. The pCO₂ values derived from SOCATv6 would therefore not represent the mean value of MARCATS like those obtained in the pCO₂ coastal-SOM-FFN vs MOM6-COBALT comparison (panel f in Fig. 3). We did however significantly strengthen our model evaluation strategy (see detailed answer in R2C2) and believe we now provide substantially more material to the reader to be convinced of the ability of our model to adequately capture the seasonality of pCO₂.**

- **R2C12: Figure 6: If any of these regions have continuous pCO2 time series in SOCATv6, SOCATv6 should also be included.**

**R2R12: We agree with the comment. This is now part of our new strategy for the model evaluation as discussed in our response to R2R2. Figure 6 was updated accordingly.**

**Supplementary correction:**

**Color bar of panel (d) in Fig. 5 has been changed for a better visibility.**

**We also replaced "model skill" by "model to coastal-SOM-FFN agreement" in the manuscript to emphasize the fact that a low agreement does not equate a poor model skill in regions with low data density.**

**Updated Fig. 5**

[revised manuscript text omitted]

---

## Author Response (AR2)

Dear Dr Roobaert and co-authors,

Thanks for the revised manuscript, which is now accepted to publication in Ocean Science. Upon submitting the final files, please consider the below technical corrections.

Response: We thank both reviewers and the editor for their constructive comments on our manuscript and are delighted that it has been accepted for publication in the Ocean Science journal. We have addressed all final comments from the editor. Please find below our detailed response.

On behalf of co-authors,

Alizée Roobaert

L225 I would not call this marginal, as the reduction is about 25%. Please change wording.

Response: We agree that this was misleading. The argument for choosing the space-varying only is based on the improvement compared to the initial bias using the traditional approach of Sarmiento and Gruber and the relatively tedious calculation of the time and space varying coefficient which is more difficult to implement. We rephrased as follows:

Line 222: "At this location, the use of the regression-based coefficients greatly improves the recovery of the simulated pCO$_2$ compared to using the traditional coefficients of Sarmiento and Gruber (2006), reducing the rmse from 83 µatm to 24 µatm corresponding to a bias reduction of 71%. The use of both space and time dependent regression-based coefficients further reduces this bias, bringing down the rmse from 24 uatm to 18 uatm corresponding to an additional 7% reduction of the initial bias (83 µatm). Based on these results, we chose to use This improvement is only differs by 25 % compared to the space-varying only, motivating our choice to use the simpler approach of the space dependent only coefficients, which is a simpler approach to implement here and in future studies. This additional improvement is however marginal, motivating our choice to use the simpler approach of the space dependent only coefficients."

L259 Not sure what you mean with: the surface pCO2 concertation (especially the last word)

Thank you for catching this typo. We corrected to "the surface pCO2 concertation" in "surface pCO2 concentration".

L275 and other places: Not sure why you call the SOCAT data raw. SOCAT data haven been QCed, and thus are not raw.

Response: We agree with the editor and removed "raw" everywhere in the text (specifically L131, L139, L279 and L313).

L352 the east coast of the U.S.A.

Response: The English guidelines of the Ocean Sciences journal states that "*Cardinal directions should only be capitalized when part of a proper noun*". The most appropriate way might actually be to use "East Coast" rather than "east coast". We looked at the literature in this region and decided to use the more commonly used regional name of "U.S. East Coast" (see for instance Najjar et al., 2018). We changed everywhere in the text and figures.

L499 and other places: I think biology should be biological processes. "Biology" is the discipline, but generally it is about the processes in this discipline which interests us.

Response: We agree with the editor and modified accordingly. We also change "physics" by physical processes. These changes are in Lines 28, 493, 503, 508, 566, and 616.

All the listed bibliography items have been updated and corrected:

L640 Bakker et al: Please check the author list, some are doubled.

Response: We updated the list of the author

L723 Change to: Global Biogeochem. Cycles, 23(1), GB1005, doi:10.1029/2008GB003349, 2009.
Response: The reference has been modified.
L726 Change to: Science, 363, 1193–1199,
Response: The reference has been modified.
L732 Please add editor, publisher and city
Response: The reference has been modified.
L746 Add volume, pages
Response: The reference has been modified.
L774 Delete n/a-n/a,
Response: The reference has been modified.
L776 Add journal name

Response: The reference has been modified. We also changed the type of citation which is a book section instead of a journal article.

L780 Change to: Science, 315, 637–639,

Response: The reference has been modified.

L806 Add volume and article number

Response: The reference has been modified.

Note that we also added some text in the Acknowledgements section and modified some typos in the text.